# Cytoplasmic dynein-1 cargo diversity is mediated by the combinatorial assembly of FTS–Hook–FHIP complexes

**Jenna R Christensen[1†], Agnieszka A Kendrick[1†], Joey B Truong[1], Adriana Aguilar-Maldonado[1], Vinit Adani[1], Monika Dzieciatkowska[2], Samara L Reck-Peterson[1,3,4]\***

[1]Department of Cellular and Molecular Medicine, San Diego, United States; [2]Department of Biochemistry and Molecular Genetics, University of Colorado Denver, Aurora, United States; [3]Howard Hughes Medical Institute, Chevy Chase, United States; [4]Division of Biological Sciences, Cell and Developmental Biology Section, San Diego, United States

**\*For correspondence:**
sreckpeterson@ucsd.edu

[†]These authors contributed equally to this work

**Abstract** In eukaryotic cells, intracellular components are organized by the microtubule motors cytoplasmic dynein-1 (dynein) and kinesins, which are linked to cargos via adaptor proteins. While ~40 kinesins transport cargo toward the plus end of microtubules, a single dynein moves cargo in the opposite direction. How dynein transports a wide variety of cargos remains an open question. The FTS–Hook–FHIP ('FHF') cargo adaptor complex links dynein to cargo in humans and fungi. As human cells have three Hooks and four FHIP proteins, we hypothesized that the combinatorial assembly of different Hook and FHIP proteins could underlie dynein cargo diversity. Using proteomic approaches, we determine the protein 'interactome' of each FHIP protein. Live-cell imaging and biochemical approaches show that different FHF complexes associate with distinct motile cargos. These complexes also move with dynein and its cofactor dynactin in single-molecule in vitro reconstitution assays. Complexes composed of FTS, FHIP1B, and Hook1/Hook3 colocalize with Rab5-tagged early endosomes via a direct interaction between FHIP1B and GTP-bound Rab5. In contrast, complexes composed of FTS, FHIP2A, and Hook2 colocalize with Rab1A-tagged ER-to-Golgi cargos and FHIP2A is involved in the motility of Rab1A tubules. Our findings suggest that combinatorial assembly of different FTS–Hook–FHIP complexes is one mechanism dynein uses to achieve cargo specificity.

## Editor's evaluation

The microtubule motor cytoplasmic dynein-1 transports diverse membrane-bound organelles, but in most cases the mechanism of cargo recognition is unknown. Christensen, Kendrick, and colleagues use BioID, in vitro assays, and live-cell fluorescence imaging to show that the three Hook family cargo adaptors form complexes of distinct composition with proteins of the FHIP family. They map how specific FHIP proteins recruit dynein to different endosome and Golgi compartments. This study provides evidence for a new mechanism through which activation of dynein motility is coupled to the selection of cargo.

## Introduction

Proper positioning of intracellular material in space and time is crucial for many cellular processes including cell division, cell signaling, and vesicle trafficking (*Burute and Kapitein, 2019*). Long distance

transport occurs primarily on polarized microtubule tracks by the motors, dynein and kinesin. While kinesin motors transport cargo predominantly toward the 'plus' end of microtubules, cytoplasmic dynein-1 ('dynein' here) transports cargo toward the 'minus' end of microtubules. In mammalian cells, kinesin and dynein are responsible for transporting many diverse cargos including membrane-bound organelles, mRNAs, and protein complexes (*Hirokawa et al., 2009*; *Reck-Peterson et al., 2018*). The expansion of the kinesin family of motors within the animal kingdom itself reflects this necessity—the presence of many distinct motors promotes the specialized delivery of many different cargos (*Kollmar and Mühlhausen, 2017*; *Miki et al., 2005*; *Welburn, 2013*). Similarly, dynein adaptors have expanded in animal cells. In many organisms, processive dynein motility requires the dynactin complex and a coiled-coil activating adaptor (*McKenney et al., 2014*; *Schlager et al., 2014*). There are ~20 known or candidate activating adaptors in human cells, several of which have been implicated in linking dynein to different cargos (*Reck-Peterson et al., 2018*; *Wang et al., 2019*). However, how activating adaptors link dynein to its cargo is only known in a few cases (*Hoogenraad et al., 2001*; *Matanis et al., 2002*; *Schlager et al., 2010*). The Hook family of activating adaptors is one of the most conserved families of dynein adaptors. Hook proteins make up one component of the 'FHF' complex consisting of FTS/AKTIP ('FTS' here), Hook, and FHIP (FHF complex subunit Hook Interacting Protein) (*Figure 1A*).

The FHF complex has been implicated in linking dynein to cargo in the filamentous fungus *Aspergillus nidulans* and human cells (*Bielska et al., 2014*; *Guo et al., 2016*; *Mattera et al., 2020*; *Xu et al., 2008*; *Yao et al., 2014*; *Zhang et al., 2014*). *A. nidulans* has one FTS (FtsA), one Hook (HookA), and one FHIP (FhipA), which together link dynein to early endosomes (*Yao et al., 2014*). Both the *A. nidulans* HookA and FtsA proteins require FhipA to associate with early endosomes (*Yao et al., 2014*), suggesting that the FHIP protein mediates cargo recognition and binding. FTS is a member of the family of inactive E2 ubiquitin-conjugating enzyme variants with varying biological functions (*Xu et al., 2008*), whose role in the FHF complex remains unclear. Human cells have one FTS, three Hooks (Hook1, Hook2, and Hook3), and four FHIPs (FHIP1A, FHIP1B, FHIP2A, and FHIP2B) (*Figure 1B*). We hypothesize that gene expansion and functional divergence of the Hook and FHIP families of proteins may result in the formation of different FHF complexes, allowing dynein to bind multiple cargos in human cells.

The exact composition of the FHF complex and whether multiple distinct FHF complexes form remains unclear. FHIP1A (also known as FHIP-L and FAM160A1) and FHIP1B (also known as FHIP, p107$^{FHIP}$, and FAM160A2) are the most well-characterized FHIP proteins. The initially described 'FHF' complex was identified by immunoprecipitation–mass spectrometry (MS) of FTS in HeLa cells (*Xu et al., 2008*), and was found to contain FHIP1B, FTS, Hook1, Hook2, and Hook3. Further studies confirmed a similar FHF complex composition (*Guo et al., 2016*) and found that FHIP1A binds FTS (*Mattera et al., 2020*), suggesting that it may also be an FHF complex component. However, very little is known about FHIP2A (also known as FAM160B1) and FHIP2B (also FAM160B2), and whether they also associate with FTS and/or Hook proteins. FHIP2A and FHIP2B were identified in Hook1 and Hook3 proteomic datasets previously generated in our laboratory (*Redwine et al., 2017*), suggesting they may also be FHF complex components.

Several studies have provided insight into the cellular roles of the FHF complex. In *A. nidulans*, the sole FHF complex links dynein to early endosomes (*Yao et al., 2014*). In human cells, the only characterized FHF complex (FTS, Hook1/2/3, and FHIP1B) has several proposed functions. This FHF complex has been shown to associate with homotypic fusion and protein sorting (HOPS) complex components and may be involved in late endosome/lysosome clustering and epidermal growth factor trafficking (*Xu et al., 2008*). In rat hippocampal neurons, an FHF complex of similar composition has been demonstrated to bind Rab5A and is involved in the retrograde axonal transport of transferrin receptor (*Guo et al., 2016*). Finally, a similar FHF complex has also been demonstrated to link dynein to the AP-4 adaptor complex (*Mattera et al., 2020*). However, very little is known about the roles of FHIP2A and FHIP2B, whether they are associated with distinct FHF complexes, and whether they are also involved in linking dynein to specific intracellular cargos.

Here, we use the FHF complex as a model cargo adaptor system to understand how dynein achieves cargo specificity. We identify and characterize the different human FHF complexes using proximity-dependent biotinylation, MS, and immunoprecipitations, and show in single-molecule motility assays using purified components that moving dynein/dynactin complexes associate with different FHF

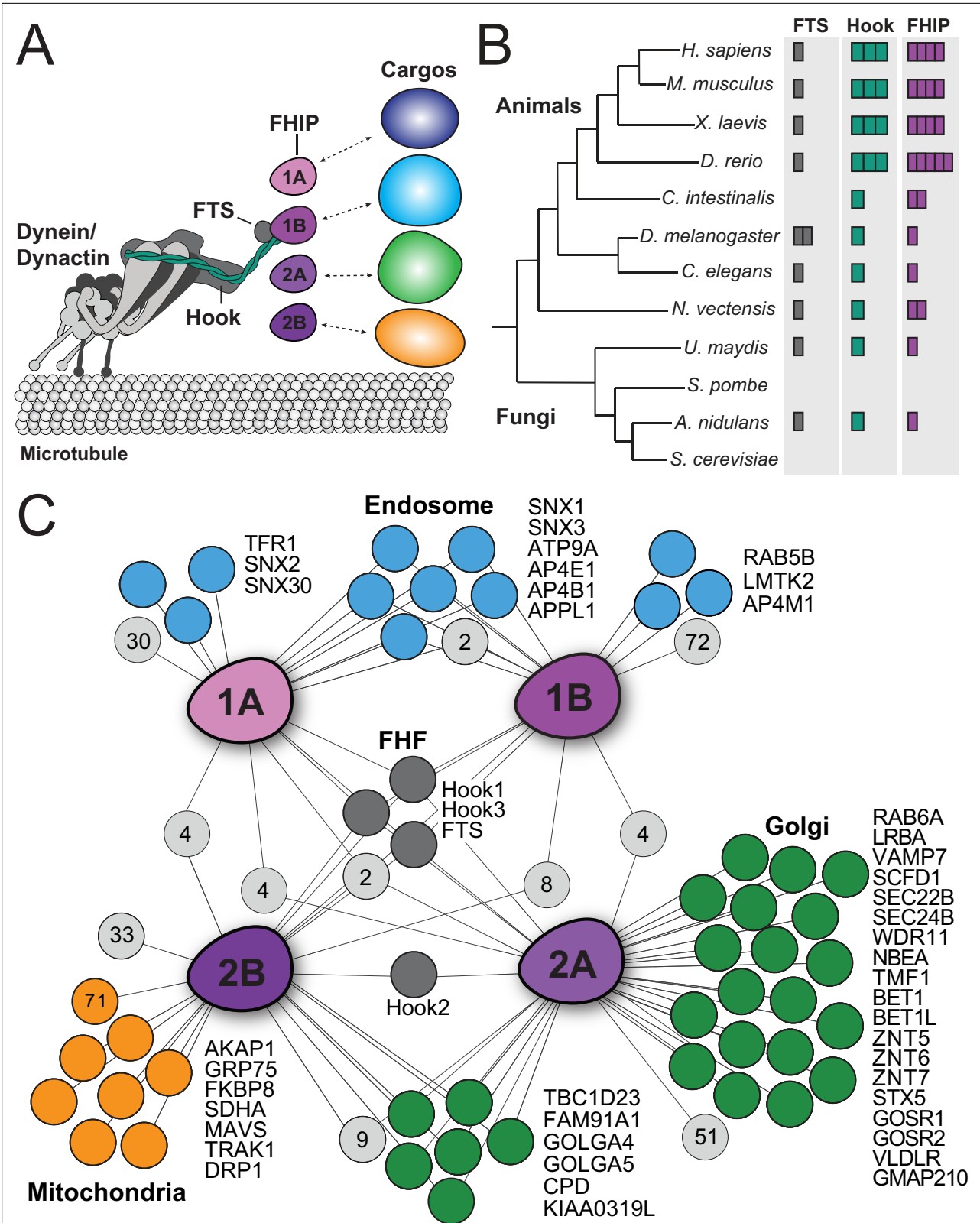

**Figure 1.** Proximity biotinylation reveals different FHIP protein interactomes. (**A**) Schematic of the dynein complex and interactions with potential cargo adaptor complexes. (**B**) Phylogenetic tree showing the number of FTS (gray), Hook (green), and FHIP (purple) protein homologs presents in each animal or fungal species listed. Each colored box denotes a putative homolog identified by reciprocal protein BLAST search. *H. sapiens*, *Homo sapiens*; *M. musculus*, *Mus musculus*; *X. laevis*, *Xenopus laevis*; *D. rerio*, *Danio rerio*; *C. intestinalis*, *Ciona intestinalis*; *D. melanogaster*, *Drosophila melanogaster*;

*Figure 1 continued*

*C. elegans, Caenorhabditis elegans*; *N. vectensis, Nematostella vectensis*; *U. maydis, Ustilago maydis*; *S. pombe, Schizosaccharomyces pombe*; *A. nidulans, Aspergillus nidulans*; *S. cerevisiae, Saccharomyces cerevisiae*. (**C**) Interaction diagram for FHIP carboxy-terminal BioID datasets. Oblong purple shapes represent FHIP1A ('1A'), FHIP1B ('1B'), FHIP2A ('2A'), and FHIP2B ('2B') datasets. Significant protein hits present in a FHIP BioID dataset that are known to be associated with specific organelles (based on gene ontology analysis) are indicated by colored circles connected by lines. Light gray circles with numbers inside represent the number of other significant protein hits in that dataset or combination of datasets. For mitochondria-associated proteins in the FHIP2B dataset, proteins known to associate with the outer mitochondrial membrane are listed by name. The orange circle with '71' denotes the number of other mitochondria-associated proteins (not outer membrane associated) found in the FHIP2B dataset. Significant hits displayed in the diagram showed a ≥threefold enrichment over the cytoplasmic BioID2 control or absence in the cytoplasmic BioID2 control, significance of p < 0.05 by Student's two-tailed *t*-test, and presence in three out of four technical replicates.

The online version of this article includes the following source data and figure supplement(s) for figure 1:

**Figure supplement 1.** Proximity biotinylation identifies different FHIP protein interactomes.

**Figure supplement 1—source data 1.** Raw uncropped immunoblot image (*Figure 1—figure supplement 1*.scn) from *Figure 1—figure supplement 1A* probed with anti-FLAG antibody.

**Figure supplement 2.** Overlap between significant hits from the different FHIP BioID datasets.

---

complexes. We also show that in cells FHF complexes containing Hook1 and/or Hook3 and FHIP1B associate with Rab5B endosomes, while complexes containing Hook2 and FHIP2A associate with Rab1A-tagged ER-to-Golgi cargos. Furthermore, FHIP1B and FHIP2A are important for the formation and movement of their corresponding cargos. Taken together, our data provide a mechanistic understanding of how a single dynein complex transports numerous cellular components.

## Results

### FHIP BioID protein interactomes

To elucidate the cellular roles of the FHIP proteins, we first identified the protein 'interactome' of each FHIP in 293T cells. To do this, we used a proximity biotinylation (BioID) approach in which the carboxy-terminus of each of the four FHIP proteins was tagged with a promiscuous biotin ligase BioID2 (*Kim et al., 2016*; *Redwine et al., 2017*; *Roux et al., 2012*). We then generated stable doxycycline-inducible 293T cell lines expressing each BioID2-tagged protein or cytoplasmic BioID2 control (*Figure 1—figure supplement 1A*). For each BioID experiment, we grew the FHIP-BioID2-expressing cell line in doxycycline and biotin-containing media for 16 hours. We then collected cells and performed a streptavidin immunoprecipitation of the biotinylated proteins and identified protein interactomes via MS. Significant protein 'hits' were determined using a label-free proteomics approach by comparison to the cytoplasmic BioID2 control (*Redwine et al., 2017*; *Zhang et al., 2010*). Proteins present in three out of four technical replicates, with threefold enrichment over the BioID2 control or not present in the BioID2 control, and with p value of less than 0.05 were considered significant.

Comparative analysis of the significant hits in each FHIP BioID dataset showed some overlapping hits between the different FHIP proteins, as well as numerous unique hits present in each FHIP dataset (*Figure 1C*, *Figure 1—figure supplement 1B*, *Supplementary file 1*). Certain components of the FHF complex (Hook1, Hook3, and FTS) were present in all four FHIP datasets (*Figure 1*), consistent with our laboratory's previous Hook1 and Hook3 BioID datasets (*Redwine et al., 2017*) and previous reports for FHIP1A and FHIP1B protein interactions (*Mattera et al., 2020*; *Xu et al., 2008*). Gene ontology (GO) analysis of significant hits in each FHIP dataset showed that different FHIP datasets had an enrichment of diverse cellular processes, functions, and components (*Figure 1C*, *Supplementary file 2*, *Eden et al., 2009*; *Eden et al., 2007*). For example, endosome-associated proteins were enriched in the FHIP1A and FHIP1B datasets (*Figure 1C* – blue circles), consistent with their previously identified endosomal functions (*Guo et al., 2016*; *Xu et al., 2008*; *Yao et al., 2014*). FHIP1B was previously proposed to interact with Rab5 (*Guo et al., 2016*) and HOPS complex components (*Xu et al., 2008*). Though we identified Rab5B as a significant hit in the FHIP1B dataset, we found no HOPS complex components in any of our FHIP datasets.

In contrast, the interactomes of FHIP2A and FHIP2B were highly enriched for Golgi-associated proteins (*Figure 1C* – green circles). Additionally, multiple proteins involved in endosome-to-trans-Golgi network (TGN) transport (TBC1D23, FAM91A1, GOLGA4, GOLGA5, and WDR11) and

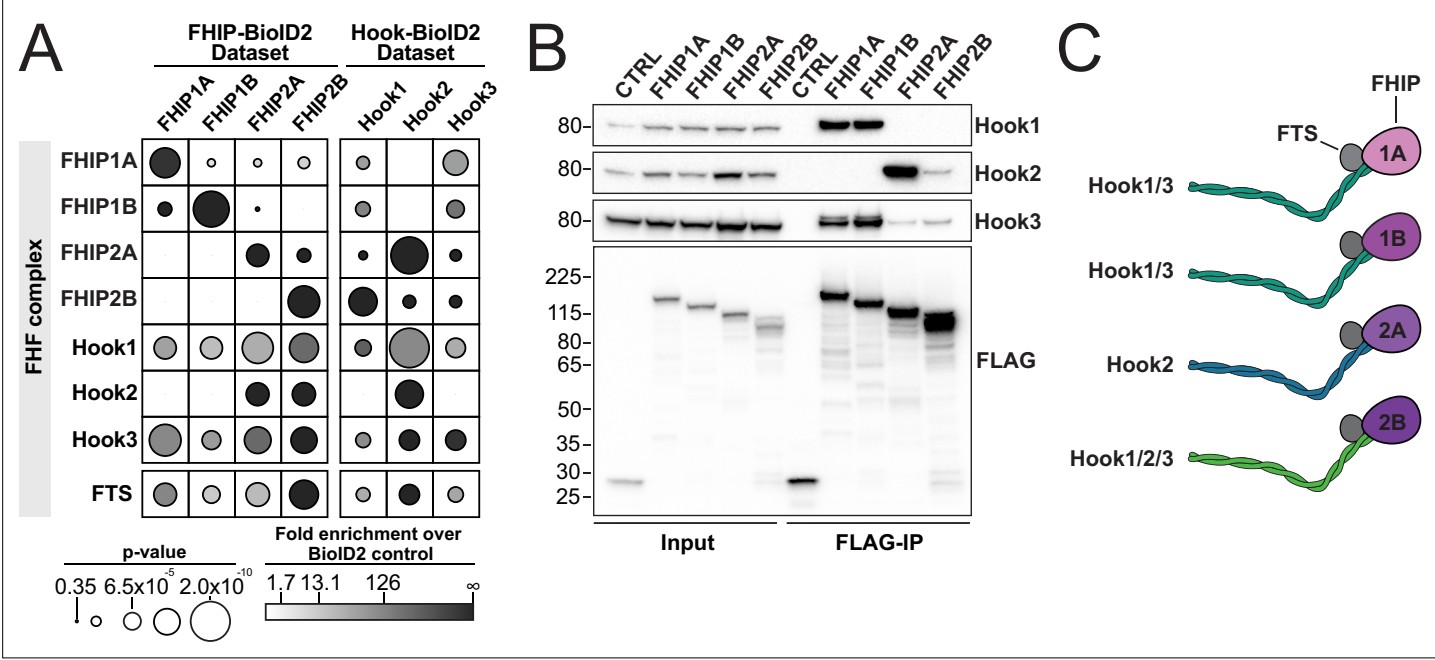

**Figure 2.** FHIP proteins preferentially interact with different Hook proteins to form different FHF complexes. (**A**) Fold-enrichment over cytoplasmic BioID2 control (grayscale intensity) and p value (circle size) for FHF complex proteins identified in carboxy-terminal FHIP and carboxy-terminal Hook BioID datasets. Hook1 and Hook3 datasets are from *Redwine et al., 2017*. (**B**) Human FHIP1A, FHIP1B, FHIP2A, and FHIP2B tagged at their carboxy termini with BioID2 and 3xFLAG were immunoprecipitated with FLAG affinity resin (FLAG-IP) from stable 293T cell lines. Immunoblots were probed with anti-Hook1, anti-Hook2, anti-Hook3, and anti-FLAG antibodies. Protein molecular weight markers are shown in kilo-Daltons to the left of each immunoblot. BioID2-3xFLAG provided a control (CTRL). Representative images from three biological replicates are shown. (**C**) Cartoon representation of the possible FHF complexes formed.

The online version of this article includes the following source data and figure supplement(s) for figure 2:

**Source data 1.** Raw uncropped immunoblot images from *Figure 2* (*Figure 2B*_Hook3.scn – anti-Hook3; Fig2B_Hook1_Hook2.scn – anti-Hook1, left side of the image and anti-Hook2, right side of the image; Fig2B_FLAG.scn – anti-FLAG) probed with the indicated antibodies.

**Figure supplement 1.** Different Hook proteins coimmunoprecipitate different FHIP proteins.

**Figure supplement 1—source data 1.** Raw uncropped immunoblot images from *Figure 2—figure supplement 1A* (Fig2supp1_FHIP1B.scn – anti-FHIP1B; Fig2supp1_FHIP2A.scn – anti-FHIP2A; Fig2supp1_FHIP2B.scn – anti-FHIP2B; Fig2supp1_FLAG.scn – anti-FLAG) probed with the indicated antibodies.

endoplasmic reticulum (ER)–Golgi transport (SCFD1, SEC22B, SEC24B, BET1, BET1L, STX5, GOSR1, and GOSR2) were found in either the FHIP2A or both the FHIP2A and FHIP2B datasets, but not in the FHIP1A or FHIP1B datasets. Although there was some overlap between the FHIP2A and FHIP2B datasets, we also observed a high level of enrichment for mitochondria-associated proteins specifically in the FHIP2B interactome (*Figure 1C* – orange circles).

We also found some cytoskeletal components enriched in the different FHIP BioID datasets. The FHIP1B dataset had multiple actomyosin-associated proteins, while the FHIP1A, FHIP2A, and FHIP2B datasets had many kinesin hits (*Figure 1—figure supplement 1C*). We did not further explore these potential kinesin interactions, but this will be an important area for future work as many cellular cargos move bidirectionally. Finally, comparative analysis of each FHIP BioID interactome with each Hook BioID interactome (*Redwine et al., 2017*) showed modest overlap (*Figure 1—figure supplement 2* and *Supplementary file 3*). Because BioID is a proximity-dependent approach this lack of substantial overlap was not surprising and supports the idea that FHIP proteins are the most cargo-proximal components of the FHF complex. Taken together, our BioID data show that different FHIP proteins associate with diverse cellular interactomes, suggesting that these proteins may link to different cellular cargos.

## Different FHIP proteins interact with different Hooks

The FHIP BioID datasets demonstrate that the FHIP proteins associate with different Hook proteins. Only Hook1 and Hook3 are present in the FHIP1A and FHIP1B BioID datasets, while all three Hooks

are present in the FHIP2A and FHIP2B datasets (*Figures 1C and 2A*). Our laboratory previously performed BioID experiments for Hook1 and Hook3 and all four FHIPs were present as significant hits in those datasets (*Redwine et al., 2017*). We performed a similar BioID experiment for Hook2 and found only FHIP2A and FHIP2B in the Hook2 BioID dataset (*Figure 2A*, *Supplementary file 1*). We confirmed these BioID findings by coimmunoprecipitations and western blotting. Our coimmunoprecipitation experiments suggest the formation of several different FHF complexes (*Figure 2C*). The first FHF complex is similar to the complex described previously (*Guo et al., 2016*; *Mattera et al., 2020*; *Xu et al., 2008*) and consists of FHIP1A/FHIP1B and Hook1/3. Supporting this, 3XFLAG-tagged FHIP1A and FHIP1B expressed in 293T cells coimmunoprecipitated Hook1 and Hook3, but not Hook2 (*Figure 2B*). In the converse experiment, Hook1 and Hook3 coimmunoprecipitated FHIP1B (*Figure 2—figure supplement 1A*). We were unable to test which Hooks coimmunoprecipitate with FHIP1A as none of the commercially available antibodies worked well in western blots.

Our data also suggest the formation of a second FHF complex consisting of FHIP2A and Hook2. 3XFLAG-tagged FHIP2A coimmunoprecipitated Hook2 and Hook3, with FHIP2A coimmunoprecipitating Hook2 to a much greater extent than Hook3 (*Figure 2B*). In the converse experiment, Hook2 coimmunoprecipitated FHIP2A (*Figure 2—figure supplement 1A*). Finally, FHIP2B appears to associate with all three Hook proteins. Though 3XFLAG-tagged FHIP2B only coimmunoprecipitated Hook2 and Hook3 (*Figure 2B*), all three Hook proteins coimmunoprecipitated FHIP2B (*Figure 2—figure supplement 1A*). Together, these data demonstrate the preferential formation of different FHF complexes: FHIP1A and FHIP1B form a complex with Hook1 and Hook3, while FHIP2A preferentially associates with Hook2, and FHIP2B is potentially capable of forming a complex with Hook1, Hook2, and Hook3 (*Figure 2C*).

## Different FHF complexes associate with motile dynein/dynactin complexes

Based on our BioID and coimmunoprecipitation experiments showing preferential formation of different FHF complexes (*Figure 2*), we next sought to determine if these FHF complexes associate with moving dynein/dynactin using in vitro reconstitution and single-molecule motility assays (*Figure 3A*). While Hook2 activates organelle motility in cells (*Dwivedi et al., 2019*), Hook2 has not been shown to activate dynein motility in vitro. Thus, we first aimed to determine if purified full-length Hook2 activates dynein/dynactin in vitro. We performed in vitro motility assays with purified dynein labeled with Alexa-TMR, unlabeled dynactin and unlabeled full-length Hook2 in the presence of Lis1 (*Figure 3—figure supplement 1A*). We included Lis1 since it increases the formation of activated dynein/dynactin complexes (*Elshenawy et al., 2020*; *Htet et al., 2020*). Doing so led to a ~threefold increase in the number of processive dynein/dynactin runs activated by full-length Hook2 (*Figure 3B* – top panel and *Figure 3—figure supplement 1B and C*). Thus, we included Lis1 in all further single-molecule experiments that involved Hook2. Dynein/dynactin complexes activated by full-length Hook2 or full-length Hook3 in the presence of Lis1 moved at similar speeds (*Figure 3—figure supplement 1D-F*). This finding shows that purified full-length Hook2 is a bona fide dynein activating adaptor.

We next sought to determine if moving dynein/dynactin complexes associate with different FHF complexes. We expressed and purified FTS-SNAP, FHIP2A-HaloTag, and coexpressed and copurified FTS and FHIP1B-HaloTag from insect cells. We labeled the FHIP proteins with Alexa-660 via the HaloTag to monitor the colocalization of these proteins with the moving TMR-dynein/dynactin complexes. Reconstitution of FTS, Hook2, and FHIP2A-Alexa 660, as well as FTS, Hook3, and FHIP1B-Alexa 660 complexes in the presence of TMR-labeled dynein and unlabeled dynactin (*Figure 3A* and *Figure 3—figure supplement 1A*) revealed the colocalization of TMR-dynein with FHIP2A-Alexa 660 (*Figure 3B* – bottom panel, yellow arrows; top panel – controls) and FHIP1B-660 (*Figure 3C* – bottom panel, yellow arrows; top panel – controls). Consistent with the role of Hook proteins in both activating dynein/dynactin complexes and linking them to FHIP/FTS, we did not observe any colocalization or motile events when Hook2 or Hook3 were omitted from the reaction mixtures (*Figure 3—figure supplement 2A, B*). The colocalization of these FHF complexes with moving dynein/dynactin had no effect or only minimal effects on dynein's motile properties including velocity, landing rates, pausing frequencies, and number of processive runs (*Figure 3D-G*, *Figure 3—figure supplement 2C-H*). Run lengths increased slightly for FTS, Hook2, FHIP2A complexes compared to complexes lacking FTS

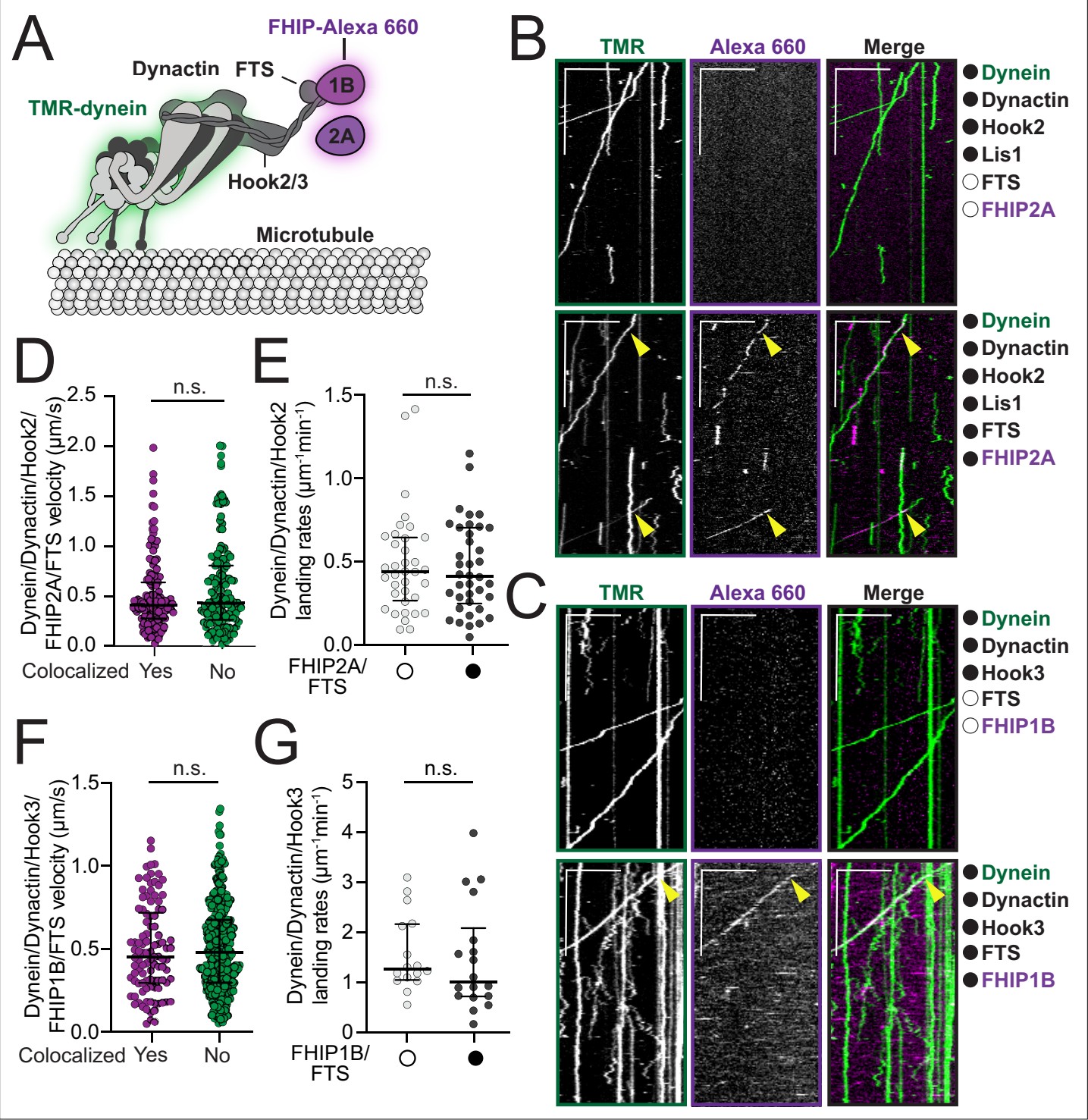

**Figure 3.** FHF complexes comigrate with moving dynein/ dynactin on microtubules. (**A**) Schematic of the dynein, dynactin, and FHF (FTS, Hook2 or Hook3, and FHIP1B or FHIP2A) complexes used in reconstitution experiments. Proteins labeled with fluorophores in motility assay are highlighted (dynein – green and FHIP1B and FHIP2A – magenta). (**B**) Representative kymographs from single-molecule motility assays with purified TMR-labeled dynein (green), unlabeled dynactin, unlabeled full-length Hook2, and unlabeled Lis1 in the absence (top panel, white circle) or presence (bottom panel, black circle) of unlabeled FTS and Alexa-660-labeled FHIP2A (magenta). Two-color colocalized runs are marked with yellow arrows on each single-channel image and in the merge. Scale bars, 10 μm (x) and 40 s (y). (**C**) Representative kymographs from single-molecule motility assays with purified TMR-labeled dynein (green), unlabeled dynactin, and unlabeled full-length Hook3 in the absence (top panel, white circle) or presence (bottom panel, black circle) of unlabeled FTS and Alexa-660-labeled -FHIP1B (magenta). A two-color colocalized run is marked with yellow arrows on each

*Figure 3 continued on next page*

Figure 3 continued

single-channel image and in the merge. Scale bars, 10 µm (*x*) and 40 s (*y*). (**D**) Single-molecule velocity (median ± interquartile range) of TMR-dynein/dynactin/Hook2 runs in the presence of FHIP2A-Alexa-660 and FTS, either colocalized with FHIP2A-Alexa-660 (yes, *n* = 140) or not (no, *n* = 150). n.s., no significance, p = 0.1193, *t*-test with Welch's correction. (**E**) Landing rates (median ± interquartile range) from TMR-dynein/dynactin/Hook2 experiments performed in the absence (white circle, *n* = 36) or in the presence of FHIP2A-Alexa-660 and FTS (black circle, *n* = 38). n.s., no significance, p = 0.9272, Mann–Whitney test. (**F**) Single-molecule velocity (median ± interquartile range) of TMR-dynein/dynactin/Hook3 runs in the presence of FHIP1B-Alexa-660 and FTS, either colocalized with FHIP1B-Alexa-660 (yes, *n* = 102) or not (no, *n* = 430). n.s., no significance, p = 0.8328, *t*-test with Welch's correction. (**G**) Landing rates (median ± interquartile range) from TMR-dynein/dynactin/Hook3 experiments performed in the absence (white circle, *n* = 16) or in the presence of FHIP1B-Alexa-660 and FTS (black circle, *n* = 18). n.s., no significance, p = 0.3653, Mann–Whitney test.

The online version of this article includes the following source data and figure supplement(s) for figure 3:

**Figure supplement 1.** Motile properties of dynein/dynactin complexes activated by full-length Hook2 and full-length Hook3.

**Figure supplement 1—source data 1.** Raw uncropped sodium dodecyl sulfate–polyacrylamide gel electrophoresis (SDS–PAGE) gel images from *Figure 3A* (*Figure 3* supp1left.scn and *Figure 3* supp1right.scn).

**Figure supplement 2.** Motile properties of dynein/dynactin complexes activated by full-length Hook2 and full-length Hook3 in the presence of FHIP and FTS proteins.

**Figure supplement 3.** Example kymographs of moving dynein/dynactin complexes in the presence of full-length Hook2, FTS and FHIP1B or full-length Hook3, FTS and FHIP2A.

and FHIP2A (*Figure 3—figure supplement 2E*). No difference in run lengths was detected for FTS, Hook3, FHIP1B complexes compared to those lacking FTS and FHIP1B (*Figure 3—figure supplement 2F*). These data show that the different FHF complexes activate and move with dynein/dynactin complexes and that no other proteins are required for their assembly.

As our BioID and coimmunoprecipitation data suggest that different FHIP proteins might associate with more than one Hook protein (*Figure 2*), we also tested if the purified FHIP proteins interact with different Hooks in single-molecule motility assays. Replacing FHIP2A with FHIP1B in the FTS/Hook2 mixture led to very rare nonmotile colocalization events (*Figure 3—figure supplement 3A*, yellow arrows). Replacement of FHIP1B with FHIP2A in the FTS/Hook3 mixture also led to very rare colocalization between motile TMR-dynein and FHIP2A-Alexa-660 (*Figure 3—figure supplement 3B*, yellow arrows). Taken together these data show that FHIP2A preferentially interacts with Hook2 and FHIP1B with Hook3 to form FHF complexes that associate with motile dynein/dynactin.

## FHIP1B and FHIP2A colocalize with microtubule-associated cargos with different morphologies

Our BioID data demonstrate that FHIP proteins have different protein interactomes. For example, the FHIP1A and FHIP1B datasets have many endosome-associated proteins, while there are many Golgi-associated proteins in the FHIP2A and FHIP2B datasets (*Figure 1C*). We chose to focus on FHIP1B and FHIP2A as representatives of these two groups of proteins (*Figure 4A*) because of their higher expression in mammalian cell lines (*Thul et al., 2017*, Human Protein Atlas, available from http://www.proteinatlas.org) and the availability of reliable western blot-compatible antibodies for these proteins. However, the FHIP1B and FHIP2A antibodies we tested did not work well for immunofluorescence experiments in our hands. Instead, we used CRISPR/Cas9 to generate FHIP1B and FHIP2A knockout cell lines (FHIP1B KO and FHIP2A KO) in human U2OS cells using two different CRISPR/Cas9-gRNAs. In each case, the appropriate FHIP was successfully knocked out without major effects on the expression of the other FHIP or the three Hook proteins (*Figure 4B*, *Figure 4—figure supplement 1A*). We then used lentiviral expression vectors to reintroduce a fluorescently tagged version of FHIP1B or FHIP2A (FHIP1B- or FHIP2A-TagRFP-T-V5) in the corresponding knockout cell line (*Figure 4C, D*, *Figure 4—figure supplement 1B*, *Video 1*). Via spinning-disk confocal imaging, fluorescently tagged FHIP1B and FHIP2A both associated with motile cargos that move on microtubules (*Figure 4E, F*, *Video 2*). However, the morphologies of these cargos were distinct. FHIP1B associated almost exclusively with punctate structures (*Figure 4C*, *Video 1*), while FHIP2A associated with both punctate and tubular structures (*Figure 4D*, *Video 1*). These different morphologies together with our BioID results suggest that FHIP1B and FHIP2A may associate with different motile cargos.

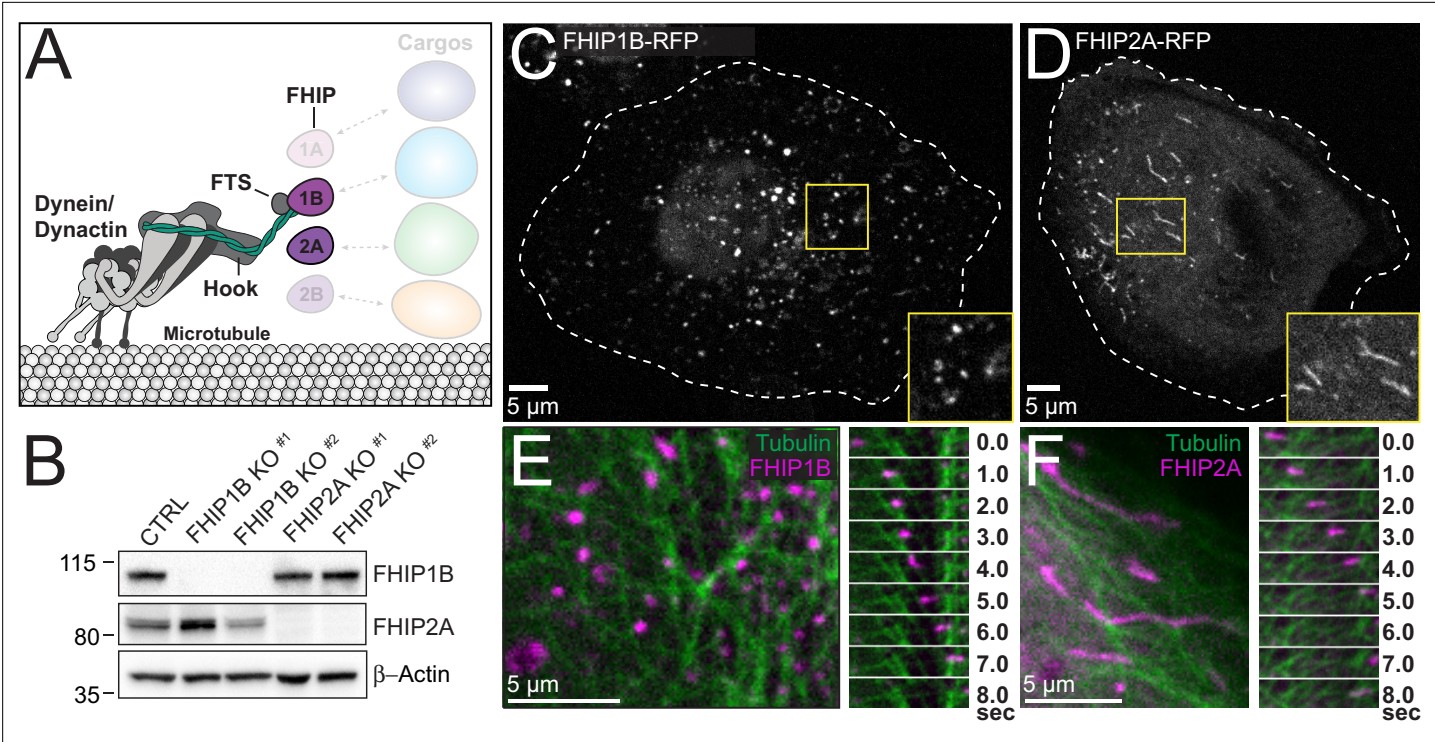

**Figure 4.** FHIP1B and FHIP2A localize to motile cargos with different morphologies. (**A**) Schematic of FHIP proteins (FHIP1B and FHIP2A) further examined in this study. (**B**) U2OS cells were transfected with control CRISPR/Cas9 (CTRL) or with two different CRISPR/Cas9-gRNAs specific for FHIP1B and FHIP2A. Knockouts were confirmed by immunoblotting with anti-FHIP1B and anti-FHIP2A antibodies. β-Actin provided a loading control. Representative images from three biological replicates are shown. Single-plane confocal micrograph of FHIP1B KO U2OS cells expressing FHIP1B-TagRFP-T-V5 (FHIP1B-RFP) (**C**) or FHIP2A KO U2OS cells expressing FHIP2A-TagRFP-T-V5 (FHIP2A-RFP) (**D**). Yellow rectangles denote region of cropped inset. Dotted lines denote cell outline. Single-plane confocal micrograph crop (left panels) and corresponding time-lapse montage (right panels) of FHIP1B KO U2OS cells expressing FHIP1B-TagRFP-T-V5 (FHIP1B) and mEmerald-tubulin (**E**) or FHIP2A KO U2OS cells expressing FHIP2A-TagRFP-T-V5 (FHIP2A) and mEmerald-tubulin (**F**).

The online version of this article includes the following source data and figure supplement(s) for figure 4:

**Source data 1.** Raw uncropped immunoblot images from *Figure 4B* (*Figure 4B*_FHIP1B.scn – anti-FHIP1B; *Figure 4B*_Actin.scn – anti-β-actin; *Figure 4B*_FHIP2A.scn– anti-FHIP2A) probed with the indicated antibodies.

**Figure supplement 1.** Protein expession levels in CRISPR/Cas9 FHIP1B and CRISPR/Cas9 FHIP2A knockout and rescue cell lines.

**Figure supplement 1—source data 1.** Raw uncropped immunoblot images from *Figure 4—figure supplement 1A*.

## FHIP1B associates with early endosomes via a direct interaction with GTP-bound Rab5B

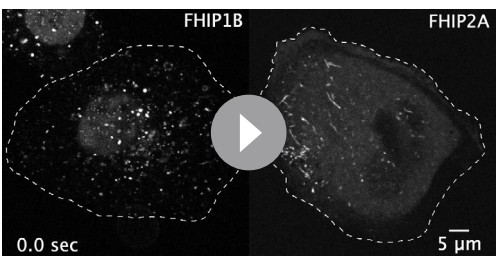

**Video 1.** FHIP1B and FHIP2A localize to motile cargos with different morphologies. Live-cell spinning-disk confocal microscopy of FHIP1B KO U2OS cells expressing FHIP1B-TagRFP-T-V5 (FHIP1B) or FHIP2A KO U2OS cells expressing FHIP2A-TagRFP-T-V5 (FHIP2A). Dotted lines denote cell outline.

https://elifesciences.org/articles/74538/figures#video1

As our FHIP1B BioID dataset had many endosome-associated proteins (*Figures 1C and 5A*), we tested whether FHIP1B was associated with early endosomes by coexpressing the early endosome marker GFP-Rab5B in our knockout cells expressing FHIP1B-TagRFP-T. We found that FHIP1B colocalized with motile and nonmotile Rab5B early endosomes via live-cell imaging (*Figure 5B, C, Video 3*). FHIP1B also colocalized with Rab5A via immunofluorescence (*Figure 5—figure supplement 1A*). Furthermore, the overexpression of FHIP1B present in our FHIP1B KO lines expressing FHIP1B-TagRFPT resulted in an accumulation of endosomes near the centrosome (*Figure 5D*), suggesting that FHIP1B may

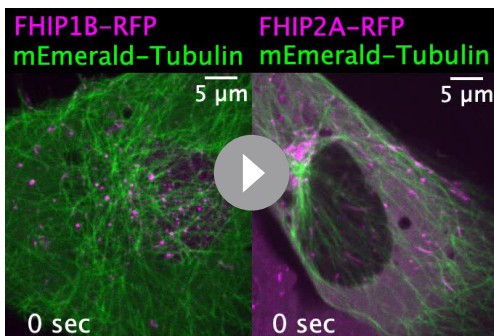

**Video 2.** FHIP1B and FHIP2A move along microtubules. Live-cell sequential acquisition spinning-disk confocal microscopy of FHIP1B KO U2OS cells expressing FHIP1B-TagRFP-T-V5 (FHIP1B-RFP, magenta) and mEmerald-Tubulin (green) or FHIP2A KO U2OS cells expressing FHIP2A-TagRFP-T-V5 (FHIP2A-RFP, magenta) and mEmerald-Tubulin (green).
https://elifesciences.org/articles/74538/figures#video2

link dynein to early endosomes, resulting in their transport to the minus ends of microtubules located near the centrosome. As FHIP1B associates with Hook1 and Hook3 via coimmunoprecipitation (*Figure 2*) and dynein/dynactin/Hook3 complexes in single-molecule motility assays (*Figure 3C* – bottom panel), we examined colocalization of all three Hook proteins with Rab5B. Using spinning-disk confocal imaging, we found that Hook1 and Hook3 colocalized with Rab5B early endosomes, while Hook2 did not (*Figure 5E*, *Videos 4–6*, controls in *Figure 5—figure supplement 1B*), suggesting that an FHF complex containing FHIP1B and Hook1 or Hook3 links dynein to Rab5B early endosomes.

Rab proteins are a family of small GTPases that undergo cycles of GTP binding and hydrolysis to regulate their cellular localization and function (*Homma et al., 2021*). GDP-bound Rabs are present in the cytosol in an inactive form, while GTP-bound Rabs associate with membranous structures and 'effector' proteins that in some cases link Rabs to motors (*Pylypenko et al., 2018*). In mammalian cells, there are three different Rab5 isoforms (Rab5A, Rab5B, and Rab5C). FHIP1B has previously been proposed to be an effector protein for Rab5A (*Guo et al., 2016*). Thus, we tested if purified FHIP1B directly interacts with purified Rab5B, as Rab5B was detected in our BioID FHIP1B dataset. To do this we mixed purified Rab5B tagged at the amino-terminus with 6xHis and carboxy-terminus with SNAP-tag with purified FHIP1B in the presence of GDP or the nonhydrolyzable GTP analog GMPPNP and performed pull-down experiments with Ni-NTA beads (*Figure 5F*). We found that GMPPNP-bound Rab5B bound FHIP1B to a higher extent as compared to GDP-bound Rab5B (*Figure 5F*), consistent with the hypothesis that FHIP1B binds to membrane-bound Rab5B.

We also tested if FHIP1B associates with other Rab GTPases. We expressed 13 GFP-tagged Rab proteins (Rab1A, 2, 3A, 4A, 5B, 6A, 7, 8, 9, 10, 11, 14, and 18) in 293T cells and performed coimmunoprecipitation experiments in the presence of the nonhydrolyzable GTP analog, GTPγS. FHIP1B coimmunoprecipitated with Rab5B, as expected, but we did not detect an interaction with any of the 12 other Rab GTPases tested (*Figure 5G*). Taken together, these data show that FHIP1B is a Rab5-specific effector.

## FHIP2A associates with Rab1A-bound ER-to-Golgi tubular intermediates

Both the FHIP2A and FHIP2B BioID datasets had many protein hits corresponding to Golgi-related processes (*Figure 1C*). Several proteins involved in endosome-to-TGN trafficking were found in both the FHIP2A and FHIP2B BioID datasets (*Figure 6—figure supplement 1A*). We therefore sought to determine whether FHIP2A was involved in endosome-to-TGN transport by expressing known GFP-tagged endosome-to-TGN transport proteins in our FHIP2A KO cells expressing FHIP2A-TagRFP-T. We found that the GFP-tagged golgin GOLGA4 colocalized with a population of FHIP2A at the Golgi apparatus (*Figure 6—figure supplement 1B*). Endosome-to-TGN transport protein TBC1D23 (*Shin et al., 2017*) colocalized with FHIP2A at the Golgi apparatus as well, but neither GOLGA4 nor TBC1D23 colocalized with motile FHIP2A tubules (*Figure 6—figure supplement 1B*), suggesting that FHIP2A was not involved in endosome-to-Golgi transport.

Many proteins known to be involved in ER-to-Golgi transport were found exclusively in the FHIP2A, but not the FHIP2B BioID dataset (*Figure 6A*). Therefore, we tested whether FHIP2A was associated with ER-to-Golgi transport. To do this, we expressed a GFP-tagged glycoprotein of vesicular stomatitis virus (VSV-G) in FHIP2A KO cells expressing FHIP2A-TagRFP-T to determine if VSV-G colocalized with FHIP2A (*Presley et al., 1997*). VSV-G is a temperature-sensitive cargo that becomes trapped in the ER when cells are grown at 40°C. Upon a temperature shift to 32°C, VSV-G is transported from the

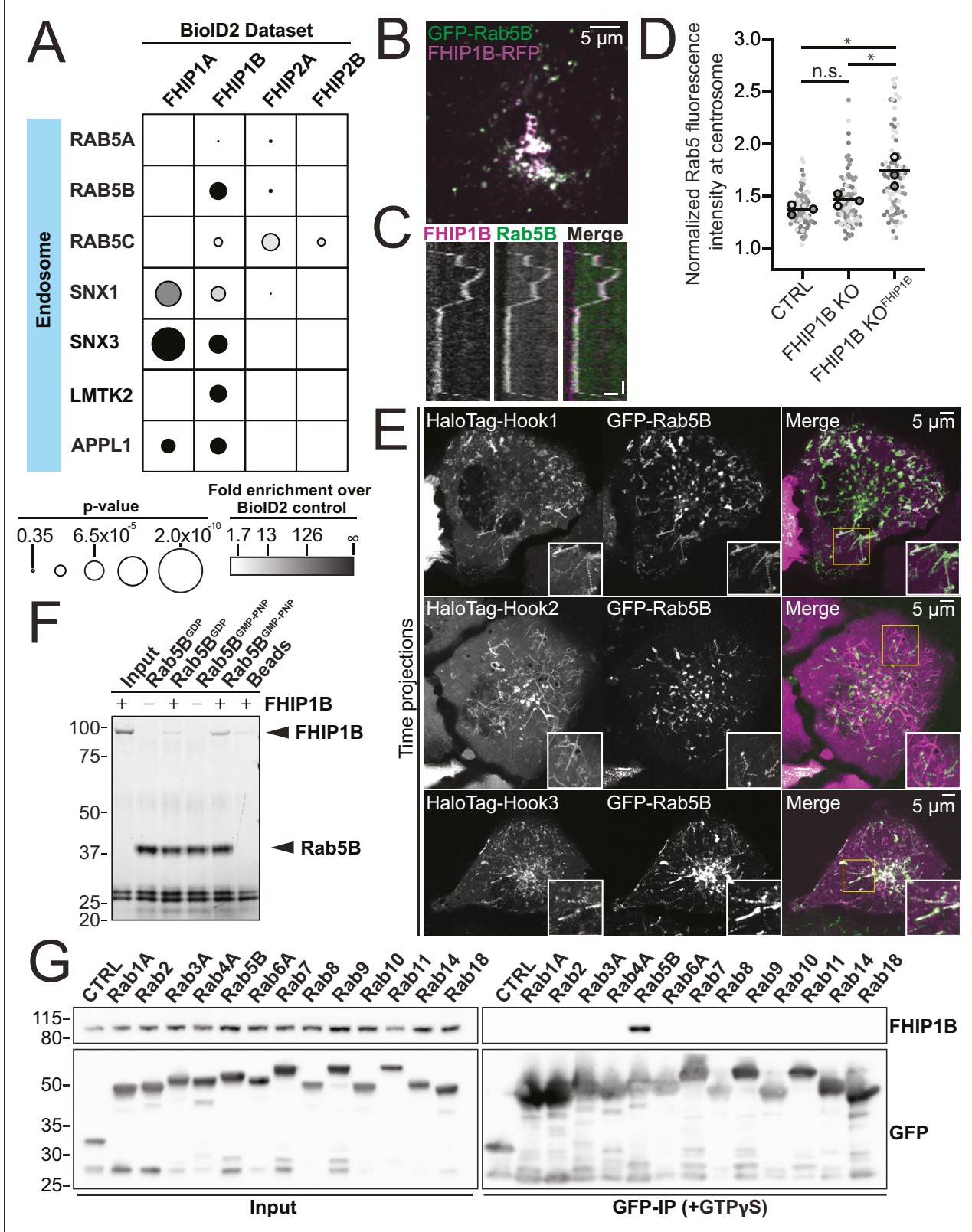

**Figure 5.** FHIP1B and Hook1/3 colocalize with early endosomes via a direct interaction between FHIP1B and GTP-bound Rab5. (**A**) Fold-enrichment over cytoplasmic BioID2 control (grayscale intensity) and p value (circle size) for selected known endosome-associated proteins identified in the indicated FHIP BioID2 datasets. (**B**) Single-plane confocal micrograph of FHIP1B KO U2OS cell expressing FHIP1B-TagRFP-T-V5 (FHIP1B-RFP) and GFP-Rab5B. (**C**) Representative kymograph of moving Rab5B puncta colocalized with FHIP1B. Scale bar, 1 μm. Time bar, 2 s. (**D**) Quantification of Rab5

*Figure 5 continued on next page*

*Figure 5 continued*

fluorescence at the centrosome in Cas9 control cells expressing cytoplasmic TagRFP-T (CTRL), FHIP1B KO cells expressing cytoplasmic TagRFP-T (FHIP1B KO), or FHIP1B KO cells expressing FHIP1B-TagRFP-T-V5 (FHIP1B KO$^{FHIP2A}$). Fluorescence intensity at the centrosome was normalized to the whole cell fluorescence, and to the areas of the regions of interest used to quantify centrosome versus whole cell fluorescence. Bold line denotes mean. Bolded circles denote means for each biological replicate. Differently shaded circles correspond to individual datapoints for cells from different biological replicates. $N$ = 77 Cas9 control, 81 FHIP1B KO, and 88 FHIP1B KO$^{FHIP1B}$. Cells from three biological replicates. n.s., no significance, p = 0.1, *p < 0.0001, Kruskal–Wallis with Dunn's post hoc test for multiple comparisons. (**E**) Single-plane time-lapse projections of U2OS cells expressing GFP-Rab5B and HaloTag-Hook1 tagged with Janelia Fluor (JF) 646 (top), HaloTag(JF646)-Hook2 (middle), or HaloTag(JF646)-Hook3. Time-lapse movies obtained by triggered acquisition. Yellow rectangle denotes region of cropped inset shown. (**F**) Purified 6His-Rab5B-SNAP preloaded with GDP or GMPPNP was bound to Ni-NTA beads and incubated with purified FHIP1B. Elutions were resolved on an sodium dodecyl sulfate–polyacrylamide gel electrophoresis (SDS–PAGE) gel. Representative image from three biological replicates is shown. (**G**) Indicated human Rab proteins tagged at the amino termini with EGFP were transiently expressed in 293T cells and immunoprecipitated with GFP nanobody affinity resin (GFP-IP) in the presence of GTPγS. Immunoblots were probed with anti-FHIP1B and anti-GFP antibodies. Protein molecular weight markers are shown in kilo-Daltons to the left of each immunoblot. 3xHA-sfGFP provided a control (CTRL). Representative images from two biological replicates are shown.

The online version of this article includes the following source data and figure supplement(s) for figure 5:

**Source data 1.** (**F**) Raw uncropped sodium dodecyl sulfate–polyacrylamide gel electrophoresis (SDS–PAGE) gel image from *Figure 5F* (Fig5F.scn) Relevant lanes are marked on the images.

**Figure supplement 1.** FHIP1B and Hook1/3 colocalize with Rab5.

ER to the Golgi and from the Golgi to the plasma membrane. We incubated cells at 40°C and after 12 hr, shifted the cells to 32°C and began imaging. We observed two primary subsets of GFP-VSV-G cargo. The first subset of cargo consisted of tubules that moved in both the anterograde and retrograde direction, and likely represent ER-to-Golgi transport intermediates (*Ben-Tekaya et al., 2005*; *Marra et al., 2001*; *Presley et al., 1997*; *Sannerud et al., 2006*). These tubules were colocalized with FHIP2A (*Figure 6B* #1, *Video 7*). The second subset of VSV-G cargo were very bright, long tubules that moved directly from the Golgi to the cell periphery (*Hirschberg et al., 1998*). These tubules represent transport from the Golgi to the plasma membrane and were not colocalized with FHIP2A (*Figure 6B* #2, *Video 7*). Therefore, our results suggest that FHIP2A is specifically associated with the ER-to-Golgi stage of VSV-G transport. We further tested whether FHIP2A was associated with ER-to-Golgi transport by expressing GFP-tagged ER-to-Golgi proteins identified in our BioID dataset in FHIP2A KO cells expressing FHIP2A-TagRFP-T. We found that FHIP2A colocalized with GMAP210, SCFD1, and BET1 (*Figure 6—figure supplement 1C*), further confirming that FHIP2A is involved in ER-to-Golgi transport.

ER-to-Golgi transport is mediated by a number of small GTPases including members of the Rab, Arf, and Arl families (*Schwaninger et al., 1992*; *Suda et al., 2017*). As FHIP1B was associated with Rab5-bound early endosomes, we sought to determine whether FHIP2A was also connected to a cargo associated via a specific small GTPase. We expressed several GFP-tagged GTPases known to be associated with the ER, Golgi, or ER-Golgi transport (Rab1A, Rab2, Rab18, Arf1, and Sar1) in our FHIP2A KO cells expressing FHIP2A-TagRFP-T to determine if FHIP2A colocalized with any of these GTPases. We found that FHIP2A strongly colocalized with GFP-Rab1A, which is crucial for several steps in ER-to-Golgi trafficking (*Saraste et al., 1995*;

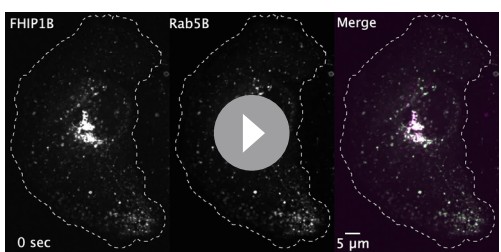

**Video 3.** FHIP1B colocalizes with early endosomes. Live-cell simultaneous acquisition spinning-disk confocal microscopy of FHIP1B KO U2OS cells expressing FHIP1B-TagRFP-T-V5 (FHIP1B, left panel, magenta in merge) and GFP-Rab5B (Rab5B, middle panel, green in merge).
https://elifesciences.org/articles/74538/figures#video3

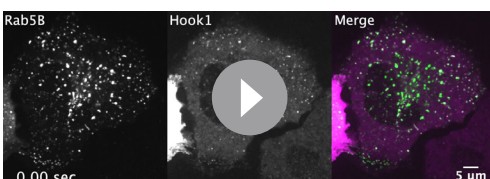

**Video 4.** Hook1 colocalizes with Rab5B early endosomes. Live-cell triggered acquisition spinning-disk confocal microscopy of U2OS cells expressing GFP-Rab5B (Rab5B, left panel, green in merge) and HaloTag-Hook1(JF646) (Hook1, middle panel, magenta in merge).
https://elifesciences.org/articles/74538/figures#video4

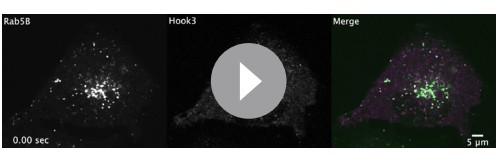

**Video 5.** Hook2 does not colocalize with Rab5B early endosomes. Live-cell triggered acquisition spinning-disk confocal microscopy of U2OS cells expressing GFP-Rab5B (Rab5B, left panel, green in merge) and HaloTag-Hook2(JF646) (Hook2, middle panel, magenta in merge).

https://elifesciences.org/articles/74538/figures#video5

*Tisdale et al., 1992*; *Westrate et al., 2020*; *Yan et al., 2021*), but none of the other small GTPases tested (*Figure 6C*, *Figure 6—figure supplement 2A*). However, FHIP2A did not immunoprecipitate with Rab1A or any of the other 12 Rabs (Rab2, 3A, 4A, 5B, 6A, 7, 8, 9, 10, 11, 14, and 18) we tested (*Figure 6—figure supplement 3A, B*), suggesting that this interaction may be indirect or require additional molecular interactions between FHIP2A and ER-to-Golgi transport intermediates.

As FHIP2A forms a complex predominantly with Hook2, but also weakly interacted with other Hook proteins (*Figure 2* and *Figure 3—figure supplement 3B*), we tested whether different Hook proteins also colocalize with Rab1A (*Videos 8–10*). We found that Hook2 showed strong colocalization with Rab1A (*Video 9*). Hook3 also showed some colocalization with Rab1A (*Video 10*), while Hook1 showed very little colocalization with Rab1A (*Figure 6D*, *Video 8*, controls in *Figure 5—figure supplement 1B*).

To test the functional relationship between Rab1A and FHIP2A, we examined the presence of motile FHIP2A tubules in cells expressing wild-type, GTP-locked (Q70L), or GDP-locked (S25N) Rab1A. In cells expressing either wild-type or GTP-locked Rab1A, FHIP2A strongly colocalized with Rab1A tubules (*Figure 7A*, *Video 11*), and at least 73% of cells had more than five motile Rab1A and FHIP2A tubules (*Figure 7B*). On the other hand, in cells expressing GDP-locked Rab1A (*Figure 7A*), 0% of cells had more than five motile Rab1A tubules, and only 22% of cells had more than five motile FHIP2A tubules (*Figure 7B*). These data suggest a link between the presence of membrane-bound Rab1A on an ER-to-Golgi tubule and the recruitment of FHIP2A.

Finally, we tested whether FHIP2A affects the formation or motility of Rab1A tubules by expressing GFP-Rab1A in CRISPR/Cas9 control, FHIP2A KO, or FHIP2A KO cells expressing FHIP2A-TagRFP-T (*Figure 7C*, *Video 12*). We found a 39% decrease in the number of motile Rab1A tubules in FHIP2A KO cells compared to control cells (*Figure 7D*). The number of motile Rab1A tubules was rescued in FHIP2A KO cells expressing FHIP2A-TagRFP-T, suggesting that FHIP2A is involved in the formation and/or motility of Rab1A tubules. We found that tubule length was slightly, but significantly, decreased in FHIP2A KO cells compared to the rescue cells (*Figure 7—figure supplement 1A*), suggesting that FHIP2A likely plays a larger role in tubule motility rather than formation. Thus, our data suggest that an FHF complex containing FHIP2A and Hook2 links dynein to Rab1A ER-to-Golgi tubular intermediates.

## Discussion

In this study, we found that functional divergence in two families of adaptor proteins, the Hook and FHIP families, allows for the formation of distinct cargo adaptor complexes that link dynein to different cellular cargos. We identified the preferential formation of specific FHF complexes (FTS/Hook1 or Hook3/FHIP1B and FTS/Hook2/FHIP2A) and found that these FHF complexes associate with moving dynein/dynactin complexes. We found that FHIP1B, Hook1, and Hook3 associate with early endosomes via a direct interaction between FHIP1B and Rab5B, while FHIP2A and Hook2 mediate transport of Rab1A-associated ER-to-Golgi transport intermediates. Together, our data demonstrate that gene expansion and functional divergence of the Hook and FHIP families of proteins provide a unique mechanism of regulating dynein's cellular cargo specificity (*Figure 8*).

**Video 6.** Hook3 colocalizes with Rab5B early endosomes. Live-cell triggered acquisition spinning-disk confocal microscopy of U2OS cells expressing GFP-Rab5B (Rab5B, left panel, green in merge) and HaloTag-Hook3(JF646) (Hook3, middle panel, magenta in merge).

https://elifesciences.org/articles/74538/figures#video6

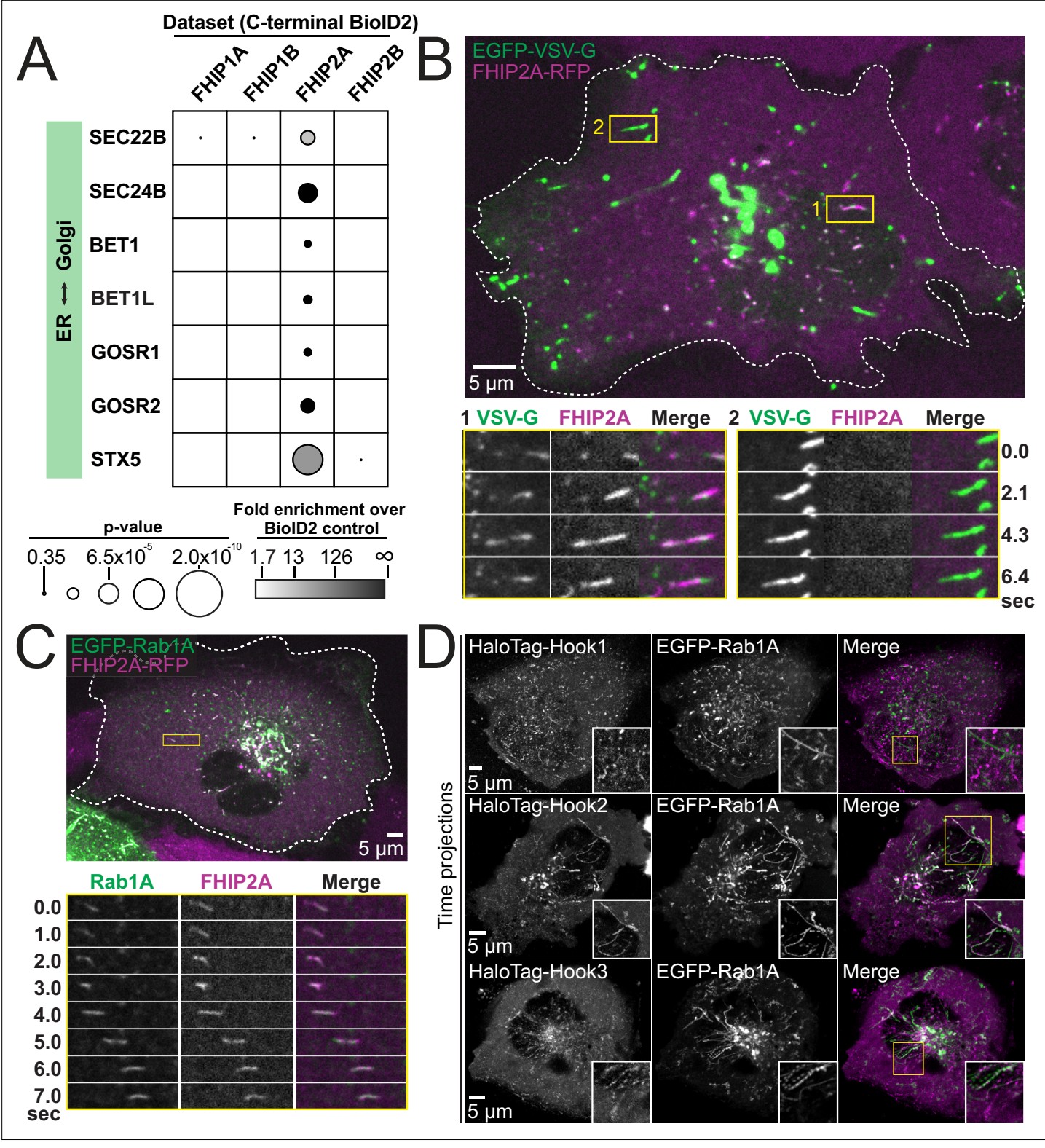

**Figure 6.** FHIP2A and Hook2 colocalize with Rab1A-associated ER-to-Golgi tubular intermediates. (**A**) Fold-enrichment over cytoplasmic BioID2 control (grayscale intensity) and p value (circle size) for selected known ER-to-Golgi proteins identified in the indicated FHIP BioID2 datasets. (**B**) Single-plane confocal micrograph of FHIP2A KO U2OS cell expressing FHIP2A-TagRFP-T-V5 (FHIP2A-RFP) and temperature-sensitive EGFP-VSV-G following temperature shift from 40°C to 32°C. Yellow rectangle (1) and corresponding time-lapse montages denote EGFP-VSV-G tubule moving from the ER to the Golgi, while (2) denotes EGFP-VSV-G tubule moving from the Golgi to the plasma membrane. (**C**) Single-plane confocal micrograph of FHIP2A KO

*Figure 6 continued on next page*

**Figure 6 continued**

U2OS cell expressing FHIP2A-TagRFP-T-V5 (FHIP2A-RFP) and EGFP-Rab1A. Dotted line denotes cell edge and yellow rectangle denotes region of time-lapse montage below. (**D**) Single-plane time-lapse projections of U2OS cells expressing EGFP-Rab1A and HaloTag-Hook1 tagged with Janelia Fluor (JF) 646 (top), HaloTag(JF646)-Hook2 (middle), or HaloTag(JF646)-Hook3. Time-lapse movies obtained by triggered acquisition. Yellow rectangle denotes region of cropped inset shown.

The online version of this article includes the following source data and figure supplement(s) for figure 6:

**Figure supplement 1.** FHIP2A is involved in ER-to-Golgi but not endosome-to-TGN transport.

**Figure supplement 2.** FHIP2A colocalizes with Rab1A.

**Figure supplement 3.** Screen for potential interactions between FHIP2A and Rab-GTPases.

**Figure supplement 3—source data 1.** (**A**) Raw uncropped immunoblot images from *Figure 6—figure supplement 3A* (Fig6supp3A_FHIP1B.scn – anti-FHIP1B; Fig6supp3A_GFP.scn– anti-GFP) probed with the indicated antibodies.

## Distinct FHF complexes link dynein to its cargos via Rab GTPases

For dynein to move processively, it requires the protein complex dynactin and a coiled-coil activating adaptor (*McKenney et al., 2014*; *Schlager et al., 2014*). There are currently ~20 known or putative activating adaptors in human cells, but it is not known how most activating adaptors link to cargo (*Reck-Peterson et al., 2018*). Rab GTPases present appealing targets for cargo-specific recognition by activating adaptors, as they are frequently the most upstream proteins denoting membrane identity (*Pfeffer, 2017*). Several activating adaptors have been implicated in linking dynein/dynactin directly or indirectly to cargo-bound Rab GTPases (*Horgan et al., 2010*; *Lee et al., 2018*; *Matanis et al., 2002*; *Schlager et al., 2010*). In this study, we found that the activating adaptors Hook1 and Hook3 link dynein to Rab5 via the intermediate protein FHIP1B, in agreement with a previous study (*Guo et al., 2016*). Here, working with pure recombinant proteins, we showed that the interaction between Rab5 and FHIP1B is direct. We also found that Hook2 links to Rab1A via FHIP2A. However, the Rab1A-FHIP2A interaction is likely indirect, as Rab1A was not present in our FHIP2A BioID dataset and we were unable to coimmunoprecipitate FHIP2A with Rab1A. Therefore, there are likely other intermediate protein(s) between FHIP2A and Rab1A that will be of interest for future studies.

## Combinatorial assembly leads to the formation of distinct FHF complexes

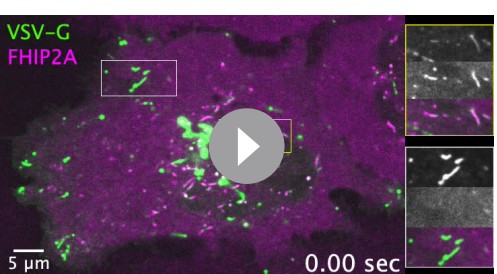

**Video 7.** FHIP2A colocalizes with a subset of VSV-G cargo. Live-cell sequential acquisition spinning-disk confocal microscopy of FHIP2A KO U2OS cell expressing FHIP2A-TagRFP-T-V5 (FHIP2A) and temperature-sensitive EGFP-VSV-G (VSV-G) following temperature shift from 40 to 32°C. In each of the two crops on righthand side, the top panel shows the VSV-G channel, the middle panel shows the FHIP2A channel, and the bottom panel shows the merge. The yellow rectangle and top crop series denote an EGFP-VSV-G tubule moving from the ER to the Golgi. The white rectangle and bottom crop series denote an EGFP-VSV-G tubule moving from the Golgi to the plasma membrane.

https://elifesciences.org/articles/74538/figures#video7

Although the initial discovery of the FHF complex suggested formation of a single complex composed of Hook proteins, FTS, and FHIP1B (*Xu et al., 2008*), our work shows that different FHF complexes exist in cells and in vitro. Specifically, we find that Hook1 and Hook3 are found in complexes containing FHIP1B, while Hook2 predominantly interacts with complexes containing FHIP2A. Our findings are further supported by studies showing that distinct Hook proteins exhibit different cellular localizations and

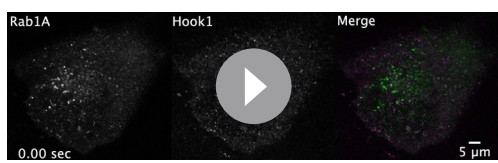

**Video 8.** Hook1 does not colocalize with Rab1A tubules. Live-cell triggered acquisition spinning-disk confocal microscopy of U2OS cells expressing EGFP-Rab1A (Rab1A, left panel, green in merge) and HaloTag-Hook1(JF646) (Hook1, middle panel, magenta in merge).

https://elifesciences.org/articles/74538/figures#video8

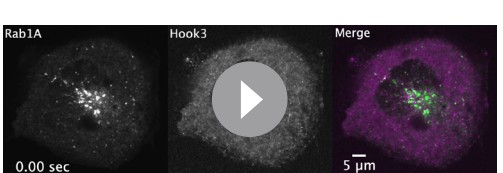

**Video 9.** Hook2 colocalizes with Rab1A tubules. Live-cell triggered acquisition spinning-disk confocal microscopy of U2OS cells expressing EGFP-Rab1A (Rab1A, left panel, green in merge) and HaloTag-Hook2(JF646) (Hook2, middle panel, magenta in merge).
https://elifesciences.org/articles/74538/figures#video9

functions, possibly due to their interactions with different FHIP proteins (*Dwivedi et al., 2019*; *Herrmann et al., 2015*; *Olenick et al., 2019*; *Xu et al., 2008*). Here, we reconstituted different dynein/dynactin/FHF complexes, showing that no other proteins are required for their formation or movement on microtubules. In our experimental conditions, the formation of these complexes did not significantly affect dynein's motile properties. Our reconstitution of complexes containing Hook2 are the first to demonstrate that Hook2 is a bona fide dynein activating adaptor. Interestingly, Hook2 seems to be more dependent on Lis1 to form activated dynein/dynactin complexes compared to complexes formed with Hook3. An important future direction will be to determine the stoichiometries of these complexes and if the addition of cargo to moving dynein/dynactin/FHF complexes affects dynein's motile properties.

FHIP proteins are likely the most cargo-proximal adaptors in the FHF complex. Here, we showed that FHIP1B directly interacts with Rab5B, which is consistent with work in *A. nidulans* where FhipA associates with RabA/Rab5-marked early endosomes in the absence of FtsA and HookA in vivo (*Yao et al., 2014*). In humans, FHIP proteins and FTS likely interact with each other in the absence of Hook proteins as yeast two-hybrid studies indicate that human FTS and FHIP1B can bind to each other independently of association with Hook (*Mattera et al., 2020*). In further support of this, mutations in a conserved carboxy-terminal α-helix in Hook1 (also conserved in Hook2 and Hook3) prevent Hook1 association with FTS and FHIP, but do not affect FTS binding to FHIP (*Xu et al., 2008*). An important area for future work will be to determine the order of assembly of Hook proteins with dynein/dynactin complexes and FHIP/FTS complexes.

## FHIP1B-containing FHF complexes link dynein to early endosomes

We demonstrate that FHIP1B is involved in dynein-mediated transport of Rab5 early endosomes. Specifically, we find that FHIP1B colocalizes with Rab5B-marked early endosomes by directly interacting with GTP-bound Rab5B. Furthermore, FHIP1B overexpression results in an accumulation of early endosomes at the centrosome. FHIP1B has been previously shown to associate with GTP-locked Rab5A via coimmunoprecipitation and yeast two-hybrid system (*Guo et al., 2016*). We also find that Hook1 and Hook3 colocalize with Rab5B-marked early endosomes, consistent with previous data for Hook1 and Hook3, and Rab5A-marked endosomes in rat hippocampal neurons (*Olenick et al., 2019*). Though we see FHIP1B association with nearly every Rab5B-marked early endosome, it is possible that FHF complexes containing FHIP1B/Hook1 may interact with a different subpopulation of endosomes from FHF complexes containing FHIP1B/Hook3. In agreement with this hypothesis, Hook1, but not Hook3 was previously shown to colocalize with a subpopulation of Rab5-marked endosomes and regulate TrkB-brain-derived neurotrophic factor (BDNF)-signaling endosomes in primary hippocampal neurons (*Olenick et al., 2019*). Together, these data suggest that FHIP1B recruits dynein to early endosomes in a Rab5 and Hook1/3-dependent manner, ultimately driving endosome movement towards the minus end of microtubules.

## FHIP2A-containing FHF complexes link dynein to ER-to-Golgi transport intermediates

Prior to this work, the role of FHIP2A in dynein-mediated cargo transport had not been investigated. We discovered that FHIP2A is important for the motility, but likely not the formation, of Rab1A-associated tubular ER-to-Golgi transport intermediates. Rab1A is involved in several steps

**Video 10.** Hook3 partially colocalizes with Rab1A tubules. Live-cell triggered acquisition spinning-disk confocal microscopy of U2OS cells expressing EGFP-Rab1A (Rab1A, left panel, green in merge) and HaloTag-Hook3(JF646) (Hook3, middle panel, magenta in merge).
https://elifesciences.org/articles/74538/figures#video10

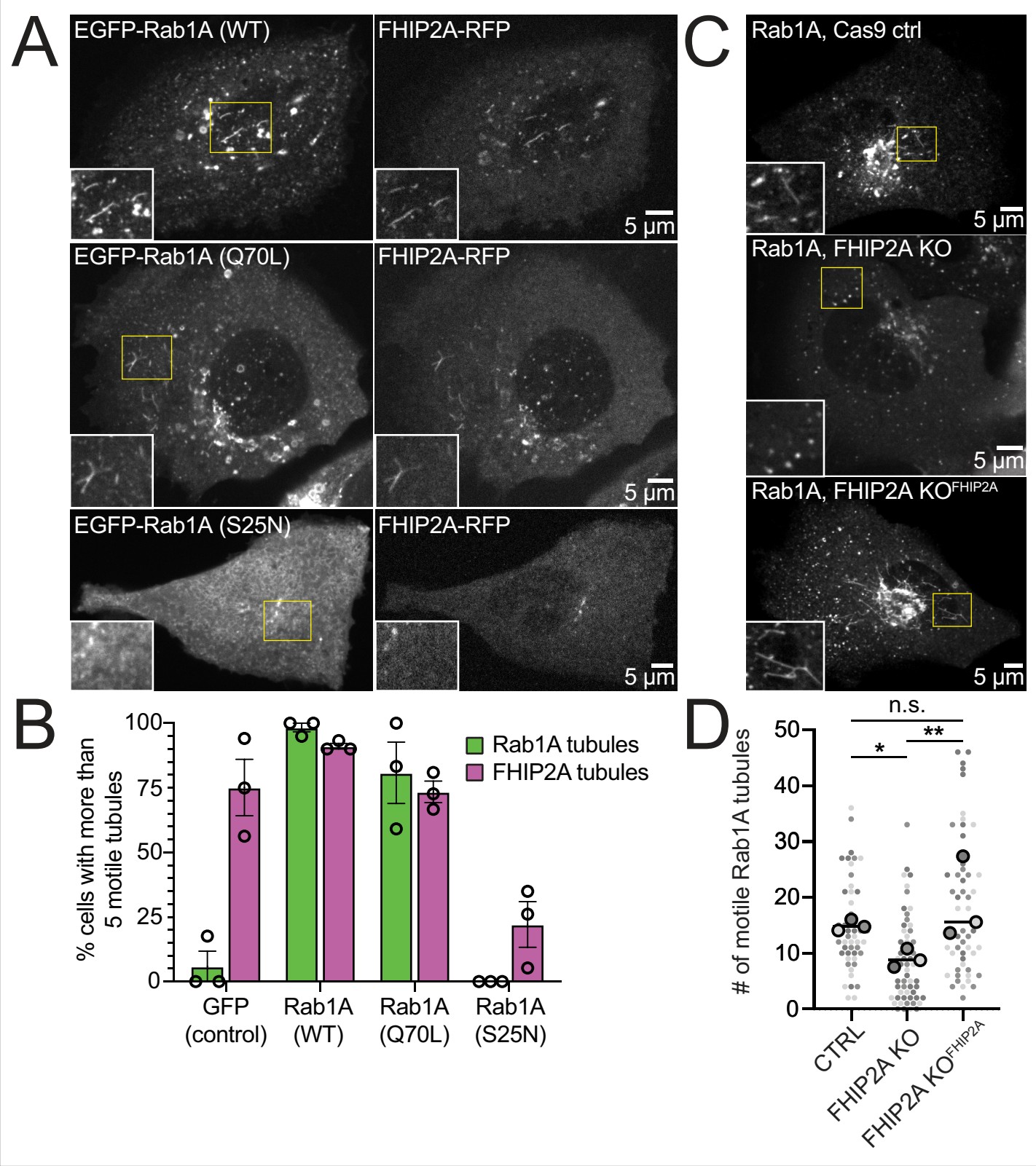

**Figure 7.** FHIP2A affects Rab1A-associated tubule formation. (**A**) Single-plane confocal micrographs of FHIP2A KO U2OS cells expressing FHIP2A-TagRFP-T-V5 (FHIP2A-RFP, right panels) and EGFP-Rab1A (WT) (top), EGFP-Rab1A (Q70L) (middle), or EGFP-Rab1A (S25N) (bottom, left panels). Yellow rectangles denote region of cropped inset. (**B**) Quantification of percentage of cells expressing more than five motile GFP/EGFP-Rab1A-positive (green) or FHIP2A-RFP positive (magenta) tubules. Error bars = mean ± SEM. $N$ = 53 GFP (control), 55 EGFP-Rab1A (WT), 61 EGFP-Rab1A (Q70L),

*Figure 7 continued on next page*

*Figure 7 continued*

and 62 EGFP-Rab1A (S25N)-expressing cells from three biological replicates. Open circles denote mean of each biological replicate. (**C**) Single-plane confocal micrographs of EGFP-Rab1A(WT) (Rab1A) in Cas9 control cells expressing cytoplasmic TagRFP-T (top), FHIP2A KO cells expressing cytoplasmic TagRFP-T (middle), or FHIP2A KO cells expressing FHIP2A-TagRFP-T-V5 (bottom). TagRFP-T panels not shown. (**D**) Quantification of the number of motile Rab1A tubules in Cas9 control cells expressing cytoplasmic TagRFP-T (Ctrl), FHIP2A KO cells expressing cytoplasmic TagRFP-T (FHIP2A KO), or FHIP2A KO cells expressing FHIP2A-TagRFP-T-V5 (FHIP2A KO[FHIP2A]). Bold line denotes median. Bolded circles denote means for each biological replicate. Differently shaded circles correspond to individual datapoints for cells from different biological replicates. $N$ = 55 Cas9 control, 59 FHIP2A KO, and 54 FHIP2A KO[FHIP2A]. Cells from three biological replicates. *p=0.0001, **p=<0.0001. Kruskal–Wallis with Dunn's post hoc test for multiple comparisons.

The online version of this article includes the following figure supplement(s) for figure 7:

**Figure supplement 1.** Motile Rab1A tubule length in control, FHIP2A knockout, and FHIP2A rescue cells.

of ER-to-Golgi transport (*Yan et al., 2021*). We show that FHIP2A colocalizes with a Rab1A-associated subset of motile VSV-G tubules, a cargo that uses microtubules and the dynein complex to move bidirectionally between the ER and Golgi (*Ben-Tekaya et al., 2005*; *Marra et al., 2001*; *Sannerud et al., 2006*; *Weigel et al., 2021*; *Yan et al., 2021*). Consistent with our data, previous work has implicated dynactin in regulating movement of ER-to-Golgi transport intermediates, as the dynactin subunit p150[Glued] colocalizes with tubular ER-to-Golgi intermediates (*Weigel et al., 2021*) and disruption of the dynactin complex via overexpression of its p50 subunit inhibits VSV-G transport (*Presley et al., 1997*). Together our findings suggest that FTS/Hook2/FHIP2A complexes link dynein/dynactin to Rab1A-marked tubular ER-to-Golgi transport intermediates.

## The protein interactomes of the different FHIP proteins suggest other candidate cargos

FHIP1A and FHIP1B share many hits in their BioID protein interactomes, suggesting partial redundancy between these proteins. For example, both FHIP1A and FHIP1B have been previously shown to associate with the AP-4 adaptor complex (*Mattera et al., 2020*), and we also found multiple AP-4 adaptor proteins in our FHIP1A and FHIP1B BioID datasets, as well as our Hook1 BioID dataset (*Redwine et al., 2017*). On the other hand, FHIP1A may also have a slightly different role from FHIP1B. For example, transferrin receptor (TFR) and two sorting nexins (SNX2 and SNX30) were found in the FHIP1A but not FHIP1B dataset, suggesting that FHIP1A may be involved in TFR recycling or another endosome-adjacent biological function using sorting nexins. A final possibility is that FHIP1A plays an important role in other cell types, since its expression is significantly lower in the 293T and U2OS cells used in this study, as compared to other FHIP proteins (*Thul et al., 2017*, Human Protein Atlas, available from http://www.proteinatlas.org).

While ER-to-Golgi transport proteins were generally found exclusively in the FHIP2A BioID datasets, FHIP2B also shared some BioID hits with FHIP2A, including several proteins known to be

**Video 11.** FHIP2A has a functional relationship with Rab1A. Live-cell triggered acquisition spinning-disk confocal microscopy of FHIP2A KO U2OS cells expressing FHIP2A-TagRFP-T-V5 (FHIP2A-TagRFPT, middle panels, magenta in merge) and EGFP-Rab1A (WT) (top), EGFP-Rab1A (Q70L) (middle), or EGFP-Rab1A (S25N) (bottom, left panels, green in merge).
https://elifesciences.org/articles/74538/figures#video11

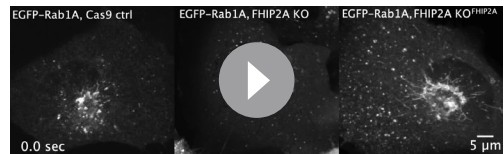

**Video 12.** FHIP2A affects Rab1A tubule motility. Live-cell spinning-disk confocal microscopy of EGFP-Rab1A(WT) in Cas9 control cells expressing cytoplasmic TagRFP-T (EGFP-Rab1A, Cas9 ctrl, left), FHIP2A KO cells expressing cytoplasmic TagRFP-T (EGFP-Rab1A, FHIP2A KO, middle), or FHIP2A KO cells expressing FHIP2A-TagRFP-T-V5 (EGFP-Rab1A, FHIP2A KO[FHIP2A], right). TagRFP-T channels not shown.
https://elifesciences.org/articles/74538/figures#video12

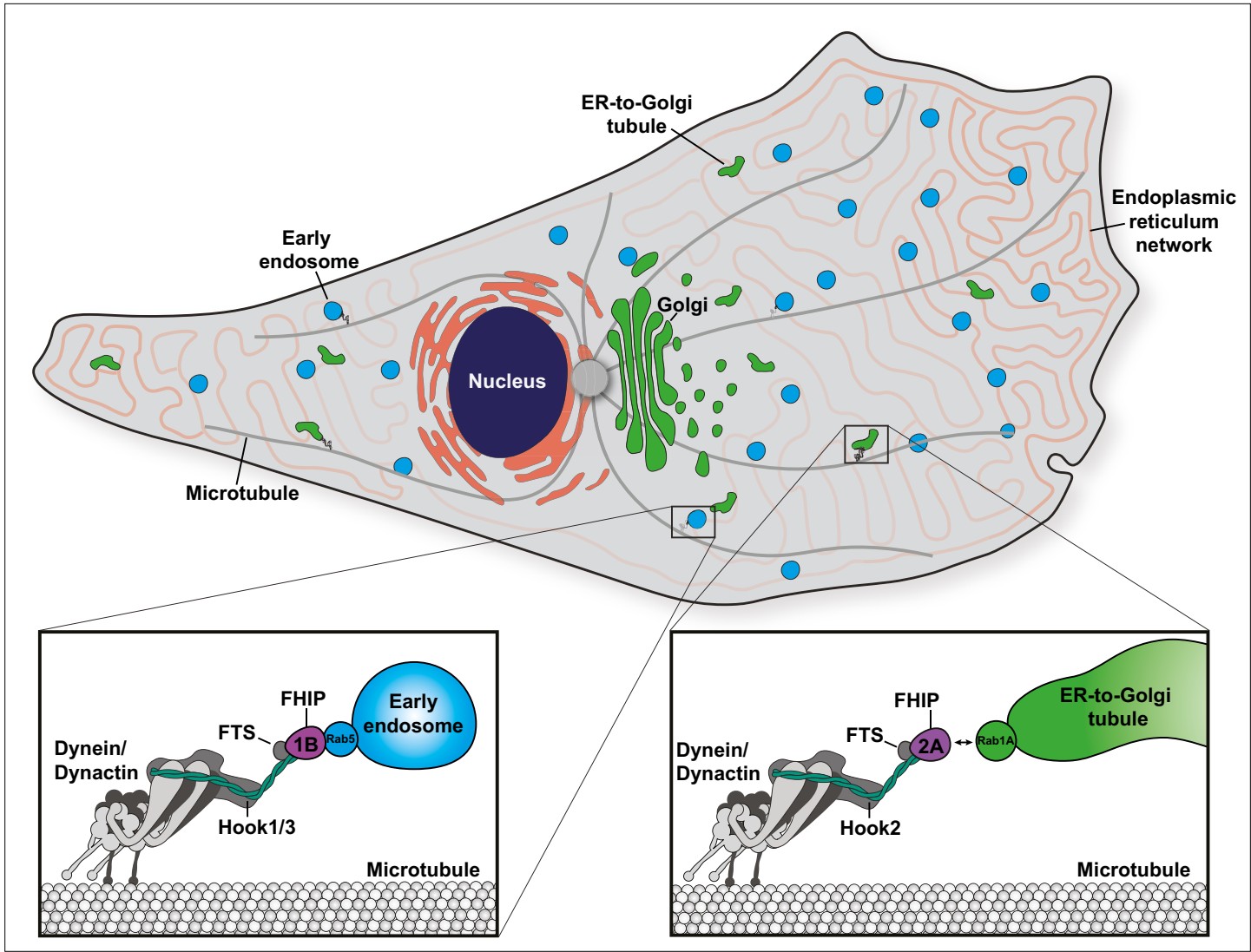

**Figure 8.** Model of the biological functions of distinct FHF complexes. Dynein/dynactin associates with each FHF complex via a direct interaction with a Hook protein. The FHF complex composed of FTS, Hook1 and/or Hook3, and FHIP1B links dynein/dynactin with early endosomes via a direct interaction between FHIP1B and Rab5. The FHF complex composed of FTS, Hook2, and FHIP2A links dynein/dynactin to Rab1A-associated ER-to-Golgi tubular transport intermediates.

involved in (TBC1D23, FAM91A1, GOLGA4, GOLGA5) or cargos of (KIAA0319L, carboxypeptidase D) endosome-to-Golgi transport (*Hirst et al., 2015*; *Shin et al., 2017*). Therefore, FHIP2B may play a role in endosome-to-Golgi transport. Additionally, a large number of mitochondria-associated proteins were found in the FHIP2B dataset, including several outer membrane proteins (AKAP1, GRP75, TRAK1, and DRP1). Thus, another possibility is that FHIP2B is involved in dynein-mediated transport of mitochondria, or another mitochondria-related function.

Finally, our FHIP1A, FHIP1B, FHIP2A, and FHIP2B protein interactomes, along with the protein interactomes of Hook1, Hook2, and Hook3 will be a powerful resource to identify additional dynein/dynactin cargos and regulatory proteins (*Redwine et al., 2017*). Further mining of these data is likely to shed additional light on how combinatorial assembly of cargo adaptor complexes can regulate dynein's cargo specificity.

# Materials and methods

## Molecular cloning

Plasmid design was performed using SnapGene software (from Insightful Science; available at snapgene.com). All plasmids used in this study, unless otherwise stated, were constructed by PCR and Gibson isothermal assembly, and are included in *Table 1*. Hook3 (clone ID: 5106726), FHIP1A (FAM160A1, clone ID: 3921921), FHIP1B (FAM160A2, clone ID: 4811942), FHIP2A (FAM160B1, clone ID: 4828347), FHIP2B (FAM160B2, clone ID: 8327615), and FTS (AKTIP, clone ID: 2967019) were obtained from Dharmacon. Hook1 (clone ID: HsCD00044030) and Hook2 isoform 2 (clone ID: HsCD00326811) cDNAs were from PlasmidID (Harvard Medical School). The 6xHis-Rab5B(A15-G191) plasmid was mutagenized in the Reck-Peterson lab to add amino acids 2–14 of human Rab5B.

## Cell lines and transfections

Human 293T and human U2OS (U-2 OS) cells were obtained from ATCC and maintained at 37°C with 5% $CO_2$ in Dulbecco's modified Eagle medium (DMEM, Corning) supplemented with 10% fetal bovine serum (FBS, Gibco) and 1% penicillin streptomycin (PenStrep, Corning). Sf9 cells were obtained from Thermo Fisher Scientific and grown in Sf-900 II SFM media (Thermo Fisher Scientific). All cells were routinely tested for mycoplasma contamination and were not authenticated after purchase.

### FLP/FRT stable cell line generation

All BioID stable cell lines were created with the FLP/FRT system and T-Rex 293T cells with constitutively expressed Tet repressor (Thermo Fisher Scientific). Stable cell lines expressing BioID-tagged genes of interest were generated by transfection with Lipofectamine 2000 (Thermo Fisher Scientific) and a combination of the appropriate pcDNA5/FRT/TO construct and Flipase expressing pOG44 plasmid. After recovery, cells were grown in DMEM containing 10% FBS, 1% PenStrep, and 50 µg/ml Hygromycin B. Gene expression was induced with 2 µg/ml doxycycline treatment for ~16 hr before cell harvest. Colonies were isolated, expanded, and screened for expression of the fusion proteins by western blotting with an anti-FLAG M2-HRP antibody (Sigma, mouse monoclonal, Cat #: A8592).

### CRISPR/Cas9-mediated genome editing

Gene editing for creation of FHIP1B and FHIP2A-depleted U2OS cells was performed as described previously (*Kendrick et al., 2019*) with slight modifications. Briefly, in vitro transcribed 20-nulceotide Alt-R crRNA along with Alt-R CRISPR-Cas9 tracrRNA were purchased from Integrated DNA Technologies (IDT). The exon 3 FHIP1B crRNA (Hs.Cas9.FAM160A2.1.AB) sequence was 5′-ACTGGACC ATGCCGCAACAGTGG-3′ and the exon 2 FHIP1B crRNA (Hs.Cas9.FAM160A2.1.AA) sequence was 5′- CTGGGCACCGTATACCTCAAGGG-3′. The exon 3 FHIP2A crRNA (Hs.Cas9.FAM160B1.1.AA) sequence was 5′-GGAATATTTGTATCGGTCACTGG-3′ and the exon 4 FHIP2A crRNA (Hs.Cas9. FAM160B1.AC) sequence was 5′-ACGTTAATGTGTGGAAGTAGTGG-3′. To prepare the Alt-R crRNA and Alt-R tracrRNA duplex, reconstituted oligos (100 µM) were mixed at equimolar concentrations in sterile PCR water and annealed at 95°C for 5 min, followed by slow cooling to room temperature. To generate knockouts 70–80% confluent U2OS cells were reverse cotransfected with 200 ng of pX459 vector and gene-specific crRNA–tracrRNA duplexes (10 nM) diluted in OptiMEM (Gibco) and combined with RNAimax reagent (Thermo Fisher Scientific). Briefly, cells were dislodged with 0.25% trypsin–EDTA and resuspended in DMEM medium. Cells were mixed with transfection complexes and plated into a 6-well plate to a final concertation of 0.64 million cells/well. Forty-eight-hour post-transfection, the cells were pulsed with 1 µg/ml puromycin for 24 hr to allow selection of pX459-transfected cells. Following puromycin selection and recovery in DMEM without puromycin, single-cell clones were plated in 96-well format by limiting dilution and cultured to allow single colonies to grow out. Clones were expanded to 12-well plates, and samples of resulting clones were screened via immunoblotting with gene-specific antibodies for FHIP1B (Abcam, FAM160A2 rabbit monoclonal, Cat #: EPR13604) and FHIP2A (Thermo Fisher Scientific, FAM160B1 rabbit polyclonal, Cat #: PA560442). A SURVEYOR mutation detection kit (IDT, #706020) was used to detect gene edited clones.

**Table 1.** Plasmids used in this study.

| Plasmid | Source or reference | Addgene # (if applicable) |
| --- | --- | --- |
| GFP-HA | This study | N/A |
| sfGFP-3XFLAG | *Kendrick et al., 2019* | N/A |
| pMX-GFP | Cell Biolabs, Cat. #RTV-050 | N/A |
| 3XHA-Rab5A | This study | N/A |
| 3XHA-Rab5A (Q79L) | This study | N/A |
| 3XHA-Rab5A (N113I) | This study | N/A |
| 3XHA-Rab5B | This study | N/A |
| 3XHA-Rab5B (Q79L) | This study | N/A |
| 3XHA-Rab5B (S34N) | This study | N/A |
| 3XHA-Rab5C | This study | N/A |
| 3XHA-Rab5C (Q80L) | This study | N/A |
| 3XHA-Rab5C (S35N) | This study | N/A |
| EGFP-Rab1A | Marci Scidmore (*Rzomp et al., 2003*) | https://www.addgene.org/49467/ |
| EGFP-Rab1A(Q70L) | Marci Scidmore (*Huang et al., 2010*) | https://www.addgene.org/49537/ |
| EGFP-Rab1A(S25N) | Marci Scidmore (*Huang et al., 2010*) | https://www.addgene.org/49539/ |
| EGFP-Rab2 | Marci Scidmore (*Huang et al., 2010*) | https://www.addgene.org/49541/ |
| EGFP-Rab3A | Marci Scidmore (*Huang et al., 2010*) | https://www.addgene.org/49542/ |
| EGFP-Rab4A | Marci Scidmore (*Rzomp et al., 2003*) | https://www.addgene.org/49434/ |
| GFP-Rab5B | Gia Voeltz (*Rowland et al., 2014*) | https://www.addgene.org/61802/ |
| EGFP-Rab6A | Marci Scidmore (*Rzomp et al., 2003*) | https://www.addgene.org/49469/ |
| EGFP-Rab7 | Richard Pagano (*Choudhury et al., 2002*) | https://www.addgene.org/12605/ |
| EGFP-Rab8 | Marci Scidmore (*Huang et al., 2010*) | https://www.addgene.org/49543/ |
| EGFP-Rab9 | Richard Pagano (*Choudhury et al., 2002*) | https://www.addgene.org/12663/ |
| EGFP-Rab10 | Marci Scidmore (*Rzomp et al., 2003*) | https://www.addgene.org/49472/ |
| EGFP-Rab11 | Richard Pagano (*Choudhury et al., 2002*) | https://www.addgene.org/12674/ |
| EGFP-Rab14 | Marci Scidmore (*Huang et al., 2010*) | https://www.addgene.org/49549/ |
| EGFP-Rab18 | Marci Scidmore (*Huang et al., 2010*) | https://www.addgene.org/49550/ |
| EGFP-Rab30 | Marci Scidmore | https://www.addgene.org/49607/ |
| Arf1-GFP | Paul Melancon (*Chun et al., 2008*) | https://www.addgene.org/39554/ |

*Table 1 continued*

| Plasmid | Source or reference | Addgene # (if applicable) |
|---|---|---|
| pDEST47-Sar1-GFP | Richard Kahn | https://www.addgene.org/67409/ |
| pcDNA5-HaloTag-3xFLAG | *Kendrick et al., 2019* | N/A |
| pcDNA5-HaloTag-Hook1-3xFLAG | This study | N/A |
| pcDNA5-HaloTag-Hook2-3xFLAG | This study | N/A |
| pcDNA5-HaloTag-Hook3-3xFLAG | *Kendrick et al., 2019* | N/A |
| mEmerald-Tubulin-6 | Michael Davidson (*Day and Davidson, 2009*) | https://www.addgene.org/54291/ |
| pEGFP-VSVG | Jennifer Lippincott-Schwartz (*Presley et al., 1997*) | https://www.addgene.org/11912/ |
| BioID G2 | Kyle Roux (*Kim et al., 2014*) | N/A |
| pLenti_CMVRO3_FHIP1B_ TagRFPT-V5 | This study | N/A |
| pLenti_CMVRO3_FHIP2A_ TagRFPT-V5 | This study | N/A |
| TBC1D23-GFP | This study | N/A |
| EGFP-GOLGA4 | This study | N/A |
| GMAP210-HA-GFP | This study | N/A |
| BET1-GFP | This study | N/A |
| SCFD1-HA-GFP | This study | N/A |
| pSpCas9(BB)–2A-Puro (PX459) V2.0 | Feng Zhang | https://www.addgene.org/62988/ |
| His-Rab5B(A15-G191) | Cheryl Arrowsmith | https://www.addgene.org/25251/ |
| 6xHis-SNAP-Rab5B(A2-G191) | This study | N/A |
| pDyn3 | *Schlager et al., 2014* | N/A |
| pcDNA5-p62-HaloTag-3xFLAG | *Redwine et al., 2017* | N/A |
| pFastBac-Hook2-HaloTag-TEV-ZZ | This study | N/A |
| pFastBac-Hook3-HaloTag-TEV-ZZ | This study | N/A |
| pBIG-ZZ-TEV-HaloTag-FHIP2A-3xFLAG-FTS | This study | N/A |
| pFastBac-FHIP1B-HaloTag-TEV-ZZ | This study | N/A |
| pLIB-3xFLAG-SNAP-FTS | This study | N/A |
| pFastBac-ZZ-TEV-Lis1 | *Baumbach et al., 2017* | N/A |
| pOPINE GFP-nanobody-6xHis | Brett Collins | https://www.addgene.org/49172/ |

## Generation of stable cell lines with fluorescently tagged FHIP1B or FHIP2A

FHIP1B KO and FHIP2A KO clones were transfected with TagRFP-T-V5 under a mutated CMV promoter (*Ferreira et al., 2011*) using a retroviral infection/MSCV-driven expression system as described previously (*Sowa et al., 2009*). Briefly, plasmid DNA (retroviral pMSCV with FHIP1B-TagRFPT-V5, FHIP2A-TagRFPT-V5 or TagRFPT-V5 genes inserted) along with viral helper constructs (retroviral MSCV-vsvg, MSCV-gag/pol) were diluted in OptiMEM (Gibco) and combined with 1 µg/µl polyethylenimine (PEI; Polysciences Inc) in a 3:1 ratio of PEI:DNA concentration. The transfection mixture was added to 293T cells, followed by incubation for 12–16 hr. Fresh DMEM was added to the cells, followed by a 24-hr incubation to allow virus production. Viral supernatant was collected, filtered, and added to recipient U2OS cells along with 1 µg/ml polybrene for infection. Stable cell lines were established by hygromycin

selection (400 µg/ml) for 48–72 hr. Expression of ectopic proteins was confirmed via western blotting with FHIP1B and FHIP2A-specific antibodies as well as anti-V5-HRP antibody (Thermo Fisher Scientific, mouse monoclonal, Cat #: R96025). After recovery, stable cell lines were grown in DMEM containing 10% FBS, 1% PenStrep, and 200 µg/ml hygromycin.

### Transient transfections

For immunoprecipitations from transiently transfected 293T cells, 1.5 × 10⁶ cells were plated onto one 10 cm dish per transfection in DMEM containing 10% FBS. Transfections were performed with PEI and 2–5 µg of transfection grade DNA (ZymoPure Plasmid MaxiPrep Kit, Zymo) per dish as described in above. Cells were harvested 48 hr after transfection. Transfection conditions for immunofluorescence and live-cell imaging are described in the Microscopy section below.

## Immunoprecipitations and immunoblotting

### Immunoprecipitations

Transiently transfected cells (see above) were collected by decanting the media and washing the cells off the dish with ice-cold phosphate-buffered saline (PBS). Cells were collected by centrifugation at 1000 × *g* for 3 min, washed again with PBS, and then transferred with PBS to Eppendorf tubes for lysis. Cells were flash frozen for storage or immediately lysed in 500 µl of lysis buffer: 50 mM 4-(2-h ydroxyethyl)-1-piperazineethanesulfonic acid (HEPES), pH 7.4; 50 mM KOAc; 2 mM MgOAc; 1 mM Ethylene glycol-bis(2-aminoethylether)-N,N,N′,N′-tetraacetic acid (EGTA); 1% (vol/vol) Triton X-100, 1 mM Dithiothreitol (DTT); and protease inhibitors (cOmplete Protease Inhibitor Cocktail, Roche) with gentle mixing at 4°C for 20 min. When indicated 2 mM GTPγS (Abcam) was included in the lysis buffer. Lysates were then centrifuged at maximum speed in a 4°C microcentrifuge for 15 min. For each immunoprecipitation, 420 µl clarified lysate was retrieved and added to 40 µl packed volume of anti-FLAG M2 agarose slurry (Sigma), Pierce anti-HA agarose slurry (Thermo Fisher Scientific) or in-house prepared anti-GFP-nanobody Sepharose slurry and incubated for ~20–24 hr at 4°C. Cells were washed four times with 1 ml of lysis buffer, and elutions were carried out by incubating beads with 60 µl of lysis buffer supplemented with NuPage LDS sample buffer (Thermo Fisher Scientific) and NuPage reducing agent (Thermo Fisher Scientific), and 10 min heat denaturation at 95°C. For 3× FLAG immunoprecipitations elutions were performed with 50 µl of lysis buffer supplemlented with 1 mg/ml 3× FLAG peptide (ApexBio).

### GFP-nanobody coupling to the beads

Purified GFP-nanobody-6His was diluted to 3 mg/ml in coupling buffer (200 mM NaHCO₃, pH 7.9; 500 mM NaCl) and incubated overnight at 4°C with NHS-activated Sepharose Fast Flow resin (Cytvia) pre-equilibrated with coupling buffer. The next day the coupling reaction was quenched by the addition of 1 M Tris–HCl, pH 7.4 (final concentration 50 mM Tris–HCl, pH 7.4) followed by a 15-min incubation in room temperature. Beads were washed with quenching buffer (50 mM Tris–HCl, pH 7.4; 150 mM NaCl) and stored in 4°C in 20% ethanol until further use.

## BioID sample preparation and MS

### Cell growth and streptavidin purification

Growth of cells and sample preparation for BioID experiments was performed as previously described (*Kendrick et al., 2019*) with slight modifications. Briefly, different BioID cell lines were plated at ~20% confluence in 15 cm dishes as four replicates, with each replicate consisting of 4 × 15 cm plates. After 24 hr, biotin was added to the media to a final concentration of 50 µM, and the cells were grown for another 16 hr. After decanting the media, cells were dislodged from each plate by pipetting with ice-cold 1× PBS. Cells were centrifuged at 1000 × *g* for 2 min, washed with ice-cold 1× PBS and the cell pellets were resuspended and lysed in 16 ml RIPA buffer 50 mM Tris–HCl, pH 8.0; 150 mM NaCl; 1% (vol/vol) NP-40, 0.5% (wt/vol) sodium deoxycholate; 0.1% (wt/vol) sodium dodecyl sulfate (SDS); 1 mM DTT; and protease inhibitors (cOmplete Protease Inhibitor Cocktail, Roche) by gentle rocking for 15 min at 4°C. The cell lysate was clarified via centrifugation at 66,000 × *g* for 30 min in a Type 70 Ti rotor (Beckman Coulter) at 4°C. The clarified lysate was retrieved and combined with prewashed 0.5 ml streptavidin-conjugated beads (Pierce Streptavidin magnetic beads) and incubated overnight at 4°C with gentle rocking. Bead/lysate mixtures were collected on a magnetic stand into a single 2 ml

round-bottom microcentrifuge tube. The beads were washed three times with 2 ml RIPA buffer and once with 1× PBS with immobilization and solution removal performed on a magnetic stand.

## On-bead digestion

Samples were prepared for MS as previously described (*Kendrick et al., 2019*). After the final wash the beads were resuspended in 100 µl of 50 mM ammonium bicarbonate (Thermo Fisher Scientific) and the proteins on the beads were reduced with 10 mM DTT for 30 min at room temperature and alkylated with 55 mM iodoacetamide (Sigma) for 30 min in the dark. Protein digestion was carried out with sequencing grade modified Trypsin (Promega) at 1/50 protease/protein (wt/wt) at 37°C overnight. After trypsin digestion, the beads were washed twice with 100 µl of 80% acetonitrile (Thermo Fisher Scientific) in 1% formic acid (Thermo Fisher Scientific) and the supernatants were collected. Samples were dried in Speed-Vac (Thermo Fisher Scientific) and desalted and concentrated on Thermo Fisher Scientific Pierce C18 Tip.

## MS data acquisition

On bead digested samples were analyzed on an Orbitrap Fusion mass spectrometer (Thermo Fisher Scientific) coupled to an Easy-nLC 1200 system (Thermo Fisher Scientific) through a nanoelectrospray ion source. Peptides were separated on a self-made C18 analytical column (100 µm internal diameter × 20 cm length) packed with 2.7 µm Cortecs particles. After equilibration with 3 µl 5% acetonitrile and 0.1% formic acid mixture, the peptides were separated by a 120 min linear gradient from 6% to 42% acetonitrile with 0.1% formic acid at 400 nl/min. LC (Optima LC/MS, Thermo Fisher Scientific) mobile phase solvents and sample dilutions were all made in 0.1% formic acid diluted in water (Buffer A) and 0.1% formic acid in 80% acetonitrile (Buffer B). Data acquisition was performed using the instrument supplied Xcalibur (version 4.1) software. Survey scans covering the mass range of 350–1800 were performed in the Orbitrap by scanning from $m/z$ 300–1800 with a resolution of 120,000 (at $m/z$ 200), an S-Lens RF Level of 30%, a maximum injection time of 50 ms, and an automatic gain control (AGC) target value of 4e5. For MS2 scan triggering, monoisotopic precursor selection was enabled, charge state filtering was limited to 2–7, an intensity threshold of 2e4 was employed, and dynamic exclusion of previously selected masses was enabled for 45 s with a tolerance of 10 ppm. MS2 scans were acquired in the Orbitrap mode with a maximum injection time of 35 ms, quadrupole isolation, an isolation window of 1.6 $m/z$, HCD collision energy of 30%, and an AGC target value of 5e4.

## MS data analysis

MS/MS spectra were extracted from raw data files and converted into.mgf files using a Proteome Discoverer Software (ver. 2.1.0.62). These .mgf files were then independently searched against human database using an in-house Mascot server (Version 2.6, Matrix Science). Mass tolerances were ±10 ppm for MS peaks, and ±25 ppm for MS/MS fragment ions. Trypsin specificity was used allowing for one missed cleavage. Methionine oxidation, protein amino-terminal acetylation, amino-terminal biotinylation, lysine biotinylation, and peptide amino-terminal pyroglutamic acid formation were all allowed as variable modifications, while carbamidomethyl of Cysteine was set as a fixed modification. Scaffold (version 4.8, Proteome Software, Portland, OR, USA) was used to validate MS/MS-based peptide and protein identifications. Peptide identifications were accepted if they could be established at greater than 95.0% probability as specified by the Peptide Prophet algorithm. Protein identifications were accepted if they could be established at greater than 99.0% probability and contained at least two identified unique peptides.

To estimate relative protein levels, Normalized Spectral Abundance Factor dNSAFs were calculated for each nonredundant protein, as described (*Redwine et al., 2017*). Average dNSAFs were calculated for each protein using replicates with nonzero dNSAF values. Enrichment of proteins in streptavidin affinity purifications from BioID-tagged stable cell lines relative to a control BioID stable cell line were calculated as the ratio of average dNSAF (ratio = avg. dNSAFORF-BioID: avg. dNSAFBioID).

BioID grayscale circle plots (*Figures 2A, 5A and 6A*, *Figure 6—figure supplement 1A*) were generated by calculating the log2(fold enrichment) against the −log10(p value), where the p value (Student's two-tailed *t*-test) was generated by comparing the replicate dNSAF values of target-BioID dataset to the BioID control. Potential interactions were considered statistically significant if they were >threefold enriched in each dataset over BioID-3X-FLAG control dataset and had p values <0.05.

The log2(fold enrichment) and −log10(p value) values were plotted for each protein in Cytoscape (*Shannon et al., 2003*). BioID interaction diagrams (*Figure 1C* and Extended Data 1d, e) were also created using Cytoscape.

## Microscopy

All immunofluorescence and live-cell imaging was performed using a Yokogawa W1 confocal scanhead mounted to a Nikon Ti2 microscope with an Apo TIRF 100 × 1.49 NA objective. The microscope was run with NIS Elements using the 488 and 561 nm lines of a six-line (405, 445, 488, 515, 561, and 640 nm) LUN-F-XL laser engine and a Prime95B camera (Photometrics). Cells were imaged in an Okolab Bold Line stage top incubator that was designed to fit in the piezo Z-stage (MadCity Labs).

### FHIP1B and FHIP2A imaging

For imaging of FHIP1B-TagRFPT/FHIP2A-TagRFPT alone or with coexpression of GFP-tagged microtubules (mEmerald-Tubulin-6), putative cargos (GFP-Rab5B, EGFP-Rab1A, EGFP-Rab1A(Q70L), EGFP-Rab1A(S25N), EGFP-GOLGA4, TBC1D23-GFP, GMAP210-GFP, BET1-GFP, SCFD1-GFP, EGFP-Rab2, EGFP-Rab18, EGFP-Rab30, Arf1-GFP, Sar1-GFP), or GFP control (pMX-GFP), 0.04 × 10$^6$ FHIP1B KO U2OS cells stably expressing FHIP1B-TagRFPT-V5 or FHIP2A KO U2OS cells stably expressing FHIP2A-TagRFPT-V5 were plated into a 6-well glass bottom plate with #1.5H coverglass (Cellvis) and grown for 24 hr. The next day, transfections were performed with Lipofectamine 2000 and 0.25–1 µg of the appropriate DNA per well. The Lipofectamine/DNA mixture was added to wells containing fresh DMEM + 10% FBS (no antibiotics) and incubated overnight. Cells were imaged 24 hr after transfection on Fluorobrite DMEM (Thermo Fisher Scientific). Single-plane images were taken from each channel every 500 ms for 1 min. Separate image channels were acquired with sequential acquisition, triggered acquisition, or simultaneous acquisition, as denoted in figure legends. For sequential acquisition, separate image channels of 488–561 were acquired using bandpass filters for each channel (525/50 and 595/50). For triggered acquisition of 488–561, firing of the 488 and 561 nm lasers was synchronized by the Prime95B trigger signal, which was integrated into a Nikon BB that itself was connected to a National Instruments 6723 DAQ board housed in an external Pxi chassis. A quad bandpass filter (Chroma ZET405/488/561/640mv2) was placed in the emission path of the W1 scanhead. For simultaneous acquisition, the emission was split with a Cairn TwinCam with a 580LP filter. The GFP emission was reflected and passed through a 514/30 BP filter onto camera 2. The TagRFP-T emission was passed through a 617/73 and an additional 600/50 filter to camera 1.

### Quantification of motile Rab1A tubules

Quantification of the number of motile Rab1A tubules was performed in FIJI (*Schindelin et al., 2012*). The 488 (EGFP-Rab1A) and 561 (TagRFPT or FHIP2A-TagRFPT-V5) channels were split and individual cells identified and isolated from the EGFP-Rab1A channel. Slice label information was removed, cells were saved as separate files, and file names were blinded using the Blind Analysis Tools macro. For each cell, the number of motile tubules present during the 1-min movie was then quantified manually using the multipoint tool. In order for a tubule to be counted, it had to move from its initial location and be more than 1 µm long at some point during its motility. Tubules that did not move from their original location and puncta shorter than 1 µm were not counted. Quantification of the percentage of cells with five or more motile tubules was performed similarly, except that cells were selected and blinded and quantifications tallied from both 488 (GFP or EGFP-Rab1A) and 561 (FHIP2A-TagRFPT) channels. Tubule length was quantified by measuring the length of each motile tubule identified in the above analysis at its longest point using FIJI. Kruskal–Wallis with Dunn's post hoc test for multiple comparisons was performed using GraphPad Prism version 9.1.2 for Mac, GraphPad Software, San Diego, CA, USA.

### HaloTag-Hook/Rab imaging

For imaging colocalization of Hook proteins with Rab1A or Rab5B, 0.02 × 10$^6$ U2OS cells were plated into a 24-well glass bottom plate with #1.5H coverglass (Cellvis) and grown for 24 hr. The next day, transfections were performed with Lipofectamine 2000 and 0.25 µg of HaloTag-3XFLAG, HaloTag-Hook1, HaloTag-Hook2, or HaloTag-Hook3 and 0.25 µg of pMX-GFP (Cell Biolabs), EGFP-Rab5B, or EGFP-Rab1A DNA per well (0.5 µg total DNA in each well). The Lipofectamine/DNA mixture was

added to wells containing fresh DMEM + 10% FBS (no antibiotics) and incubated overnight. The next day, Fluorobrite DMEM with 25 nM HaloTag-JF646 ligand (*Grimm et al., 2017*) in DMSO was added to the wells for 15 min under normal incubation conditions. Following incubation with dye, cells were rinsed once with warm Fluorobrite DMEM and imaged in fresh Fluorobrite DMEM. Single-plane images were taken every 500 ms for 30 s. Separate image channels were acquired with triggered acquisition of 488–640, firing of the 488 and 640 nm lasers was synchronized and filtered as described in 'FHIP1B and FHIP2A imaging'. Time-lapse projections were generated by taking maximum intensity projections of the time-lapse movie in FIJI.

### VSV-G imaging

For imaging of FHIP2A/VSV-G colocalization, $0.075 \times 10^6$ FHIP2A KO U2OS cells stably expressing FHIP2A-TagRFPT-V5 were plated into a 35 mm glass bottom FluoroDish with 23 mm well (World Precision Instruments) and grown for 24 hr. The next day, transfections were performed with Lipofectamine 2000 and 0.5 µg EGFP-VSV-G DNA per well. The Lipofectamine/DNA mixture was added to wells containing fresh DMEM+10% FBS (no antibiotics). After 12 hr, cells were shifted to 40°C for 12 hr to retain EGFP-VSVG in the ER. Following incubation, cells were switched to Fluorobrite media. Immediately before imaging, cells were transferred to the microscope with Okolab chamber mounted and set to 40°C. The temperature was then reduced to 32°C to induce transport of EGFP-VSVG from the ER and imaging was initiated. Separate image channels were acquired sequentially using bandpass filters for each channel (525/50 and 595/50). Single-plane images were taken from each channel approximately every second for 1 min.

## Immunofluorescence

### FHIP1B/Rab5B immunofluorescence

For imaging of Rab5B accumulation surrounding the centrosome, $0.015 \times 10^6$ U2OS cells (Cas9 ctrl, FHIP1B KO, or FHIP1B KO stably expressing FHIP1B-TagRFPT-V5) were grown on fibronectin-coated glass coverslips and fixed with a solution of 3.7% formaldehyde (Electron Microscopy Sciences), 90% methanol (VWR), and 5 mM Sodium Bicarbonate (VWR Chemicals) in 1× PBS. Cells were washed with PBS then permeabilized and blocked with 5% normal goat serum (Cell Signaling Technology) and 1% bovine serum albumin (BSA) in PBS containing 0.3% Triton X-100 (Sigma). Cells were immunostained for 1 hr at room temperature with mouse monoclonal anti-γ-tubulin (Abcam, Cat #: 27074) and rabbit monoclonal anti-Rab5 (Cell Signaling, Cat #: 3547) diluted in PBS with 1% BSA (Sigma) and 0.1% Triton X-100. Cells were washed with PBS and stained with goat anti-mouse IgG (H + L) Alexa Fluor-488 (Thermo Fisher Scientific Cat. No. A11001, 1:500 dilution) and goat anti-rabbit IgG (H + L) Alexa Fluor-568 (Thermo Fisher Cat. No. A11036, 1:500 dilution). Cells were then washed with PBS with calcium and magnesium (Thermo Fisher Scientific) and stained with iFluor 647 Wheat Germ Agglutinin Conjugate (AAT Bioquest) and DAPI. Cells and coverslips were mounted on glass slides with Prolong Glass Antifade Mountant (Thermo Scientific Scientific). Z-stacks were acquired using a piezo Z stage (Mad City Labs). Separate image channels were acquired sequentially using bandpass filters for each channel (405: 455/50, 488: 525/50, 561: 595/50, 640: 705/75).

To quantify early endosome accumulation at the centrosome, sum projections of Z-stacks were created in FIJI for each separate channel. Channels were separated and the brightest γ-tubulin puncta was identified in the 488 channel as the centrosome and a 60 pixel-wide circle was drawn around it, creating a region of interest (ROI). A whole cell ROI was then created from the 640 channel using wheat germ agglutinin as a cell membrane marker. Each ROI was then applied to the Rab5/561 channel to quantify the fluorescence intensity at the centrosome and of the whole cell. The fluorescence intensity at the centrosome divided by the whole cell fluorescence was then quantified to determine the percentage of total fluorescence present at the centrosome. Then, the area of the centrosome ROI was divided by the area of the whole cell ROI to determine the percentage of the area of the cell that the centrosome ROI included. The fluorescence intensity ratio was then divided by the area ratio and plotted using GraphPad Prism. Kruskal–Wallis with Dunn's post hoc test for multiple comparisons was performed using GraphPad Prism.

For imaging of colocalization between FHIP1B and Rab5, FHIP1B KO cells stably expressing FHIP1B-TagRFPT-V5 were grown, plated, fixed, and stained as described above, but with mouse

monoclonal anti-V5 (Thermo Fisher Scientific, Cat #: R960-25) and rabbit monoclonal anti-Rab5 (Cell Signaling, Cat #: 3547) primary antibodies.

## Protein expression and Purifications

### Baculovirus generation from Sf9 insect cells

Full-length SNAP-dynein, Lis1, Hook2-HaloTag, Hook3-HaloTag, HaloTag-FHIP1B-FTS, FHIP2A-HaloTag, and FTS-SNAP constructs were expressed in Sf9 cells as described previously (*Baumbach et al., 2017*; *Htet et al., 2020*; *Schlager et al., 2014*). Briefly, the pDyn3 plasmid containing the human dynein genes, pFastBac, pLIB, or pBIG plasmids containing the above indicated proteins (also listed in *Table 1*) were transformed into DH10EmBacY chemically competent cells with heat shock at 42°C for 15 s followed by incubation at 37°C for 5 hr in S.O.C media (Thermo Fisher Scientific). The cells were then plated on LB-agar plates containing kanamycin (50 µg/ml), gentamicin (7 µg/ml), tetracycline (10 µg/ml), BluoGal (100 µg/ml) and IPTG (40 µg/ml) and positive clones were identified by a blue/white color screen after 48 hr. For full-length human dynein constructs, white colonies were additionally tested for the presence of all six dynein genes using PCR. These colonies were then grown overnight in LB medium containing kanamycin (50 µg/ml), gentamicin (7 µg/ml) and tetracycline (10 µg/ml) at 37°C. Bacmid DNA was extracted from overnight cultures using an isopropanol precipitation method as described previously (*Zhang et al., 2017*). 2 ml of Sf9 cells at $0.5 \times 10^6$ cells/ml were transfected with 2 µg of fresh bacmid DNA and FuGene HD transfection reagent (Promega) at a 3:1 transfection reagent to DNA ratio according to the manufacturer's instructions. After 3 days, the supernatant containing the 'V0' virus was harvested by centrifugation at $200 \times g$ for 5 min at 4°C. To generate 'V1', 1 ml of the V0 virus was used to transfect 50 ml of Sf9 cells at $1 \times 10^6$ cells/ml. After 3 days, the supernatant containing the V1 virus was harvested by centrifugation at $200 \times g$ for 5 min at 4°C and stored in the dark at 4°C until use. For protein expression, 4–6 ml of the V1 virus were used to transfect 400–600 ml of Sf9 cells at $1 \times 10^6$ cells/ml. After 3 days, the cells were harvested by centrifugation at $3000 \times g$ for 10 min at 4°C. The pellet was resuspended in 10 ml of ice-cold PBS and pelleted again. The pellet was flash frozen in liquid nitrogen and stored at −80°C until further purification.

### Dynein

Full-length dynein was purified as described previously (*Schlager et al., 2014*). Frozen cell pellets from a 400 ml culture were resuspended in 40 ml of Dynein-lysis buffer (50 mM HEPES [pH 7.4], 100 mM sodium chloride, 1 mM DTT, 0.1 mM Mg-ATP, 0.5 mM Pefabloc, 10% [vol/vol] glycerol) supplemented with 1 cOmplete EDTA-free protease inhibitor cocktail tablet (Roche) per 50 ml and lysed using a Dounce homogenizer (10 strokes with a loose plunger and 15 strokes with a tight plunger). The lysate was clarified by centrifuging at $183,960 \times g$ for 88 min in Type 70 Ti rotor (Beckman). The clarified supernatant was incubated with 4 ml of IgG Sepharose 6 Fast Flow beads (GE Healthcare Life Sciences) for 3–4 hr on a roller. The beads were transferred to a gravity flow column, washed with 200 ml of Dynein-lysis buffer and 300 ml of TEV buffer (50 mM Tris–HCl [pH 8.0], 250 mM potassium acetate, 2 mM magnesium acetate, 1 mM EGTA, 1 mM DTT, 0.1 mM Mg-ATP, 10% [vol/vol] glycerol). For fluorescent labeling of carboxy-terminal SNAPf tag, dynein-coated beads were labeled with 5 µM SNAP-Cell-TMR (New England Biolabs) in the column for 10 min at room temperature and unbound dye was removed with a 300 ml wash with TEV buffer at 4°C. The beads were then resuspended and incubated in 15 ml of TEV buffer supplemented with 0.5 mM Pefabloc and 0.2 mg/ml TEV protease (purified in the Reck-Peterson lab) overnight on a roller. The supernatant containing cleaved proteins was concentrated using a 100K MWCO concentrator (EMD Millipore) to 500 µl and purified via size exclusion chromatography on a TSKgel G4000SWXL column (TOSOH Bioscience) with GF150 buffer (25 mM HEPES [pH7.4], 150 mM potassium chloride, 1 mM magnesium chloride, 5 mM DTT, 0.1 mM Mg-ATP) at 1 ml/min. The peak fractions were collected, buffer exchanged into a GF150 buffer supplemented with 10% glycerol, concentrated to 0.1–0.5 mg/ml using a 100K MWCO concentrator (EMD Millipore). Purity was evaluated on sodium dodecyl sulfate–polyacrylamide gel electrophoresis (SDS–PAGE) gels and protein aliquots were snap frozen in liquid N2 and stored at −80°C.

### Lis1

Lis1 constructs were purified from frozen cell pellets from 400 ml culture as described previously (*Baumbach et al., 2017*). Lysis and clarification steps were similar to full-length dynein purification

except lysis was performed in Lis1-lysis buffer (30 mM HEPES [pH 7.4], 50 mM potassium acetate, 2 mM magnesium acetate, 1 mM EGTA, 300 mM potassium chloride, 1 mM DTT, 0.5 mM Pefabloc, 10% [vol/vol] glycerol) supplemented with 1 cOmplete EDTA-free protease inhibitor cocktail tablet (Roche) per 50 ml was used. The clarified supernatant was incubated with 0.5 ml of IgG Sepharose 6 Fast Flow beads (GE Healthcare Life Sciences) for 2–3 hr on a roller. The beads were transferred to a gravity flow column, washed with 20 ml of Lis1-lysis buffer, 100 ml of modified TEV buffer (10 mM Tris–HCl [pH 8.0], 2 mM magnesium acetate, 150 mM potassium acetate, 1 mM EGTA, 1 mM DTT, 10% [vol/vol] glycerol) supplemented with 100 mM potassium acetate, and 50 ml of modified TEV buffer. Lis1 was cleaved from IgG beads via incubation with 0.2 mg/ml TEV protease overnight on a roller. The cleaved Lis1 was filtered by centrifuging with an Ultrafree-MC VV filter (EMD Millipore) in a tabletop centrifuge. Purity was evaluated on SDS–PAGE gels and protein aliquots were snap frozen in liquid N2 and stored at −80 °C.

## FHIP1B, FHIP2A, Hook2, and Hook3

FHIP1B (FHIP1B-HaloTag-TEV-ZZ), FHIP2A (ZZ-TEV-HaloTag-FHIP2A), FHIP2A-FTS (ZZ-TEV-HaloTag-FHIP2A-3xFLAG-FTS), Hook2 (Hook2-HaloTag-Tev-ZZ), and Hook3 (Hook3-HaloTag-Tev-ZZ) were purified from Baculovirus-infected Sf9 insect cells as described for the dynein and Lis1 purifications with slight modifications. Cell pellets from 600 ml cultures were resuspended in lysis buffer (50 mM Tris–HCl, pH 7.4; 300 mM NaCl) supplemented with 1× protease inhibitor cocktail (cOmplete Protease Inhibitor Cocktail, Roche), 1 mM DTT and 0.5 mM Pefabloc SC (Sigma-Aldrich). For fluorescent labeling of HaloTag, FHIP1B or FHIP2A-coated IgG beads were labeled with 5 µM HaloTag-Alexa-660 (New England Biolabs) in the column for 10 min at room temperature and unbound dye was removed with a 300 ml wash with TEV buffer at 4°C. After overnight protein cleavage from IgG beads with TEV protease the cleaved protein eluate was loaded onto a Mono Q 5/50GL 1 ml column on an AKTA FPLC (Cytvia). The column was washed with 8 ml of Buffer A (50 mM Tris–HCl, pH 8.0; 1 mM DTT) and then subjected to a 15 ml linear gradient from 10–55% Buffer B mixed with Buffer A (Buffer B: 50 mM Tris–HCl, pH 8.0; 1 M NaCl; 1 mM DTT), followed by 5 ml additional 100% Buffer B. Fractions containing pure protein were pooled and buffer exchanged through iterative rounds of dilution and concentration on a 50 or 100 kDa MWCO centrifugal filter (Amicon Ultra, Millipore) into final storage buffer (25 mM HEPES, pH 7.4; 50 mM KCl, 1 mM MgCl₂, 1 mM DTT, 10% glycerol). Purity was evaluated on SDS–PAGE gels and protein aliquots were snap frozen in liquid N2 and stored at −80°C.

## FTS

FHIP1B (3XFLAG-SNAP-FTS) was purified from Baculovirus-infected Sf9 insect cells as described for the dynein and Lis1 purifications with slight modifications. Cell pellets from 400 ml cultures were resuspended in lysis buffer (50 mM Tris–HCl, pH 7.4; 300 mM NaCl) supplemented with 1× protease inhibitor cocktail (cOmplete Protease Inhibitor Cocktail, Roche), 1 mM DTT and 0.5 mM Pefabloc SC (Sigma-Aldrich). The clarified lysate was incubated with 1.5 ml packed anti-FLAG M2 agarose resin (Sigma) at 4°C for 16 hr. After incubation, the lysate/resin mixture was centrifuged at 1000 × g for 2 min at 4°C to pellet the resin and the supernatant was decanted. The resin was transferred to a column at 4°C and the column was washed with 100 ml low salt wash buffer (50 mM Tris–HCl, pH 7.4; 50 mM NaCl; 1 mM DTT), 100 ml high salt wash buffer (50 mM Tris–HCl, pH 7.4; 300 mM NaCl; 1 mM DTT; 0.02% Triton X-100), and finally with 50 ml of low salt wash buffer. The resin was resuspended in 800 µl of low salt wash buffer containing 2 mg/ml 3X-FLAG peptide (ApexBio) and incubated for 30 min at 4°C. The mixture was retrieved and centrifuged through a small filter column to remove the resin. The eluate was next loaded onto a Mono Q 5/50GL 1 ml column on an AKTA FPLC (GE Healthcare). The column was washed with 5 ml of Buffer A (50 mM Tris–HCl, pH 8.0; 1 mM DTT) and then subjected to a 15 ml linear gradient from 5% to 45% Buffer B mixed with Buffer A (Buffer B = 50 mM Tris–HCl, pH 8.0; 1 M NaCl; 1 mM DTT), followed by 5 ml additional 100% Buffer B. Fractions containing pure FTS (~35–40% Buffer B) were pooled and buffer exchanged through iterative rounds of dilution and concentration on a 30 kDa MWCO centrifugal filter (Amicon Ultra, Millipore) using GF150 buffer with 10% glycerol. For copurification of FHIP1B and FTS, both proteins were coexpressed in Baculovirus-infected Sf9 insect cells as described above. After overnight protein cleavage from IgG beads with TEV protease the cleaved protein eluate was further purified via size exclusion chromatography on a Superdex 200 column (Cytvia) with modified GF150 buffer (25 mM

HEPES [pH 7.4], 150 mM potassium chloride, 1 mM DTT) at 1 ml/min. The peak fractions containing Alexa-660-HaloTag-FHIP1B coeluted with SNAP-FLAG were pooled and buffer exchanged through iterative rounds of dilution and concentration on a 100 kDa MWCO centrifugal filter (Amicon Ultra, Millipore) into final storage buffer (25 mM HEPES, pH 7.4; 150 mM potassium chloride, 1 mM DTT, 10% glycerol). Purity was evaluated on SDS–PAGE gels and protein aliquots were snap frozen in liquid N2 and stored at −80°C.

## Dynactin

Dynactin (p62-HaloTag-3XFLAG) was purified from a stable 293T cell line as previously described (*Redwine et al., 2017*). Briefly, frozen pellets from 293T cells (160 × 15 cm plates) were resuspended in DLB lysis buffer (30 mM HEPES, pH 7.4; 50 mM KOAc; 2 mM MgOAc; 1 mM EGTA, pH 7.5; 10% glycerol) supplemented with 0.5 mM ATP, 0.2% Triton X-100 and 1× protease inhibitor cocktail (cOmplete Protease Inhibitor Cocktail, Roche) and gently mixed at 4°C for 15 min. The lysed cells were then centrifuged at 66,000 × rpm in a Ti70 rotor (Beckman) at 4°C for 30 min. The clarified lysate was retrieved and added to 3 ml packed anti-FLAG M2 agarose resin (Sigma) and incubated with gentle mixing at 4°C for 16 hr. After incubation, the lysate/resin mixture was centrifuged at 1000 × *g* for 2 min at 4°C to pellet the resin and the supernatant was decanted. The resin was transferred to a column at 4°C and the column was washed with 100 ml low salt wash buffer (30 mM HEPES, pH 7.4; 50 mM KOAc; 2 mM MgOAc; 1 mM EGTA, pH 7.5; 10% glycerol; 1 mM DTT; 0.5 mM ATP; 0.5 mM Pefabloc; 0.02% Triton X-100), 100 ml high salt wash buffer (30 mM HEPES, pH 7.4; 250 mM KOAc; 2 mM MgOAc; 1 mM EGTA, pH 7.5; 10% glycerol; 1 mM DTT; 0.5 mM ATP; 0.5 mM Pefabloc; 0.02% Triton X-100), and finally with 50 ml of low salt wash buffer. The resin was resuspended in 800 µl of low salt wash buffer containing 2 mg/ml 3X-FLAG peptide (ApexBio) and incubated for 30 min at 4°C. The mixture was retrieved and centrifuged through a small filter column to remove the resin. The eluate was then loaded onto a Mono Q 5/50GL 1 ml column on an AKTA FPLC (GE Healthcare). The column was washed with 5 ml of Buffer A (50 mM Tris–HCl, pH 8.0; 2 mM MgOAc; 1 mM EGTA; 1 mM DTT) and then subjected to a 26 ml linear gradient from 35% to 100% Buffer B mixed with Buffer A (Buffer B = 50 mM Tris–HCl, pH 8.0; 1 M KOAc; 2 mM MgOAc; 1 mM EGTA; 1 mM DTT), followed by 5 ml additional 100% Buffer B. Fractions containing pure dynactin (~75–80% Buffer B) were pooled and buffer exchanged through iterative rounds of dilution and concentration on a 100 kDa MWCO centrifugal filter (Amicon Ultra, Millipore) using GF150 buffer with 10% glycerol. Purity was evaluated on SDS–PAGE gels and protein aliquots were snap frozen in liquid N2 and stored at −80°C.

## Rab5B

6xHis-SNAP-Rab5B(A2-G191) was transformed into BL21-CodonPlus (DE3)-RIPL cells (Agilent). 2 l of cells were grown at 37°C in LB media to a 600 nm optical density of 0.4–0.8 before the temperature was reduced to 18°C and expression was induced with 0.5 mM IPTG. After 16–18 hr, the cells were harvested via centrifugation for 6 min at 4°C at 6000 × *g* in a Beckman-Coulter JLA 8.1000 fixed angle rotor. Pellets were resuspended in 40 ml of Ni Buffer A (50 mM phosphate, pH 7.4; 500 mM NaCl; 20 mM imidazole) supplemented with 0.5 mM Pefabloc SC (Sigma-Aldrich) and 1× protease inhibitor cocktail (cOmplete Protease Inhibitor Cocktail, Roche). Cells were lysed via sonication (Branson Digital Sonifier) and clarified via centrifugation at 66,000 × *g* for 30 min in a Type 70 Ti rotor (Beckman) at 4°C. The supernatant was loaded onto a HisTrap 5/50 1 ml column on an AKTA FPLC (Cytvia). The column was washed with 10 ml of Ni Buffer A and then subjected to a 15 ml linear gradient from 10% to 55% Buffer B mixed with Buffer A (Buffer B: 50 mM phosphate, pH 7.4; 500 mM NaCl; 1 M imidazole), followed by 5 ml additional 100% Buffer B. Fractions containing 6xHis-SNAP-Rab5B(A2-G191) were pooled, concentrated using a 10 kDa MWCO centrifugal filter (Amicon Ultra, Millipore) and further purified using MonoQ Column as described for FHIP1B purification above. Fractions containing pure 6xHis-SNAP-Rab5B(A2-G191) (~30% Buffer B) were pooled and buffer exchanged through iterative rounds of dilution and concentration on a 10 kDa MWCO centrifugal filter (Amicon Ultra, Millipore) into final storage buffer (50 mM Tris–HCl, pH 7.4; 150 mM NaCl, 5 mM MgCl$_2$, 1 mM DTT, 10% glycerol). Purity was evaluated on SDS–PAGE gels and protein aliquots were snap frozen in liquid N2 and stored at −80°C.

### GFP nanobody

GFP-nanobody-6xHis was transformed into BL21-CodonPlus (DE3)-RIPL cells (Agilent) and protein expression and purification were performed as described above for Rab5B with slight modifications. His-Trap fractions containing GFP-nanobody-6xHis were pooled and concentrated using a 10 kDa MWCO centrifugal filter (Amicon Ultra, Millipore). Concentrated GFP-nanobody-6xHis was applied to Superdex S75 Increase 10/300 GL Column connected to an AKTA FPLC (Cytvia) and run in a buffer containing 25 mM HEPES, pH 7.4; 50 mM KCl; 1 mM MgCl$_2$; 1 mM DTT. Peak fractions containing pure GFP-nanobody-6xHis were pooled, concentrated and buffer exchanged to the same buffer supplemented with 10% glycerol using a 10 kDa MWCO centrifugal filter (Amicon Ultra, Millipore). Aliquots were snap frozen in LN2 and stored at −80°C. Protein purity was checked on a Sypro (Thermo Fisher Scientific) stained SDS–PAGE gel.

## TIRF microscopy

Imaging was performed with an inverted microscope (Nikon, Ti-E Eclipse) equipped with a 100 × 1.49 N.A. oil immersion objective (Nikon, Plano Apo). The *xy* position of the stage was controlled by ProScan linear motor stage controller (Prior). The microscope was equipped with an MLC400B laser launch (Agilent) equipped with 405 nm (30 mW), 488 nm (90 mW), 561 nm (90 mW), and 640 nm (170 mW) laser lines. The excitation and emission paths were filtered using appropriate single band-pass filter cubes (Chroma). The emitted signals were detected with an electron multiplying CCD camera (Andor Technology, iXon Ultra 888). Illumination and image acquisition were controlled by NIS Elements Advanced Research software (Nikon).

### Single-molecule motility assays

Single-molecule motility assays were performed in flow chambers assembled as described previously (*Case et al., 1997*) using the TIRF microscopy setup described above. Biotinylated and PEGylated coverslips (microsurfaces) were used to reduce nonspecific binding. Microtubules were polymerized from tubulin prepared from bovine brain as previously described (*Waterman-Storer, 2001*). Microtubules contained ~10% biotin-tubulin for attachment to streptavidin-coated cover slip and ~10% Alexa Fluor 488 (Thermo Fisher Scientific) tubulin for visualization. Imaging buffer was DLB supplemented with 20 µM taxol, 1 mg/ml casein, 1 mM Mg-ATP, 71.5 mM βME (beta mercaptoethanol) and an oxygen scavenger system, 0.4% glucose, 45 µg/ml glucose catalase (Sigma-Aldrich), and 1.15 mg/ml glucose oxidase (Sigma-Aldrich). Images were recorded every 0.4 s for 3 min. Movies showing significant drift were not analyzed.

All movies were collected in two-color setup with the following protein concentrations: 125 pM TMR-dynein, 250 pM unlabeled dynactin, 9 nM Hook3 or Hook2, 450 nM FTS, 450 nM FHIP1B (FHIP1B-HaloTag-Alexa660), or 450 nM FHIP2A (FHIP2A-HaloTag-Alexa660). All experiments performed with Hook2, unless otherwise stated, also included 15 nM Lis1. For conditions missing one of the indicated protein components, corresponding matching buffers were used. The FHF complexes composed of FTS, Hook2 or Hook3, and FHIP1B or FHIP2A samples were incubated on ice for 10 min prior to dynein and dynactin addition. Each protein mixture was then incubated on ice for an additional 5 min following Lis1 addition and an additional 5 min incubation on ice prior to TIRF imaging.

### TIRF data analysis

The velocity of moving particles was calculated form kymographs generated in ImageJ as described previously (*Roberts et al., 2014*). Velocities were only calculated from molecules that moved processively for greater than five frames. Nonmotile or diffusive events were not considered in velocity calculations. Processive events were defined as events that move unidirectionally and do not exhibit directional changes greater than 600 nm. Diffusive events were defined as events that exhibit at least one bidirectional movement greater than 600 nm in each direction. Single-molecule movements that change apparent behavior (e.g., shift from nonmotile to processive) were considered as multivelocity events and counted as multiple events. For run length analysis, the length of each track in a multive-locity event was combined to calculate the total run length. Landing rates were calculated by counting the number of processive events that start after the first frame and end before the last frame of each movie and dividing this number by the microtubule length and total movie time. Pausing frequencies were calculated by counting the total number of pauses in multivelocity events and dividing this

number by the total run length. Colocalization events were manually scored in kymographs. Two-color partially colocalized processive events were counted as colocalized events.

Data visualization and statistical analyses were performed using GraphPad Prism (9.2; GraphPad Software), Excel (version 16.52; Microsoft), and ImageJ (2.0). Brightness and contrast were adjusted in Image J for all videos and kymographs. The exact value of *n*, evaluation of statistical significance, and the specific statistical analysis are described in the corresponding figures and figure legends. All experiments were analyzed from at least three independent replicates and cumulative data are shown in each figure, unless otherwise indicated.

### In vitro binding assay

Direct binding between Rab5B and FHIP1B was assessed by incubating 15 µM 6xHis-SNAP-Rab5B with 25-molar excess of GMP-PNP (Abcam) or GPD (Abcam) in nucleotide buffer (50 mM Tris–HCl, pH 7.5, 300 mM NaCl, 10 mM EDTA) overnight at 4°C. The next day, reactions were supplemented with 15 mM $MgCl_2$ and nucleotide-bound 6xHis-SNAP-Rab5B was diluted to 5 µM in Ni-binding buffer (50 mM Tris–HCl, pH 7.5, 500 mM NaCl, 20 mM imidazole, 0.1% casein). 6xHis-SNAP-Rab5B was then captured onto 20 µl Dynabeads His-Tag Magnetic Beads (Thermo Fisher Scientific) and washed three times in Ni-binding buffer in 2 ml Protein Lo Bind Tubes (Eppendorf). Bead conjugated 6xHis-SNAP-Rab5B was incubated for 15 min in room temperature with gentle agitation. The beads were then washed three times with Ni-binding buffer and 250 nM FHIP1B-HaloTag diluted in modified Ni-binding buffer (25 mM Tris–HCl, pH 7.5, 250 mM NaCl, 10 mM imidazole, 0.1% casein) was added to the beads. FHIP1B-HaloTag-conjugated bead complexes were gently shaken for 30 min at room temperature following three washes with 1 ml Ni-binding buffer. FHIP1B-HaloTag was eluted of the beads by applying 30 µl of elution buffer (50 mM Tris–HCl, pH 7.5, 150 mM NaCl, 300 mM imidazole, 0.01% NP-40) and 10-min incubation at room temperature. The elution was analyzed via SDS–PAGE on a gel stained with SYPRO Red (Thermo Fisher Scientific).

### Homolog identification

FTS/AKTIP (FTS), Hook, and FHIP homologs in *Figure 1B* were identified by reciprocal protein BLAST (BLASTp) search (*Altschul et al., 1990*). FTS (Uniprot accession: Q9H8T0), Hook1 (UniProt accession: Q9UJC3), Hook2 (Q96ED9), Hook3 (Q86VS8), FHIP1A (Q05DH4), FHIP1B (Q8N612), FHIP2A (Q5W0V3), or FHIP2B (Q86V87) protein sequences from *Homo sapiens* were retrieved from UniProtKB. Each protein sequence was used as a BLASTp query using the following BLASTp algorithm parameters (BLOSUM 62 matrix, word size 3, filtering low complexity regions) within the reference protein (refseq_protein) organism databases for *Mus musculus* (BLAST taxid: 10090), *Xenopus laevis* (taxid: 8355), *Danio rerio* (taxid: 7955), *Ciona intestinalis* (taxid: 7719), *Drosophila melanogaster* (taxid: 7227), *Caenorhabditis elegans* (taxid: 6239), *Nematostella vectensis* (taxid: 45351), *Ustilago maydis* (taxid: 5270), *Schizosaccharomyces pombe* (taxid: 4896), *A. nidulans* (taxid: 162425), and *Saccharomyces cerevisiae* (taxid: 4932). A follow-up PSI-BLAST search using the same parameters and *Drosophila melanogaster* hook (Uniprot accession: Q24185) as a query and three iterations of max 500 sequences were also performed. From each organism, identified proteins with an e-value of less than $1.0 \times 10^{-5}$ were identified. Each of these hits was then used as a BLASTp query against the *Homo sapiens* database (taxid: 9606) to determine if the corresponding *Homo sapiens* protein was the top reciprocal hit. Homologs found on both *X. laevis* S- and L- subgenomes were only counted as a single homolog. Any identified protein in which the corresponding *Homo sapiens* protein was also the top reciprocal hit was counted as a putative homolog on *Figure 1B* and *Figure 1—figure supplement 1A*. These searches did not identify *A. nidulans* FhipA (gene *AN10801*), but this protein has been previously identified and characterized as a FHIP homolog (*Yao et al., 2014*).

## Acknowledgements

We thank Eric Griffis, Hiroyuki Hakozaki, and the Nikon Imaging Center at UC San Diego for assistance with live-cell imaging and quantification. JRC is funded by a MOSAIC K99/R00 award from the National Institutes of Health (K99GM140269). AAK is supported by American Cancer Society PF-18-190-01-CCG. SLRP is supported by the Howard Hughes Medical Institute and NIH grants R01GM121772 and 1R35GM141825.

## Additional information

### Competing interests

Samara L Reck-Peterson: Reviewing editor, eLife. The other authors declare that no competing interests exist.

### Funding

| Funder | Grant reference number | Author |
|---|---|---|
| Howard Hughes Medical Institute | | Samara L Reck-Peterson |
| National Institutes of Health | R01GM121772 | Samara L Reck-Peterson |
| National Institutes of Health | R35GM141825 | Samara L Reck-Peterson |
| National Institutes of Health | K99GM140269 | Jenna R Christensen |
| American Cancer Society | PF-18-190-01-CCG | Agnieszka A Kendrick |

The funders had no role in study design, data collection, and interpretation, or the decision to submit the work for publication.

### Author contributions

Jenna R Christensen, Agnieszka A Kendrick, Conceptualization, Formal analysis, Funding acquisition, Investigation, Methodology, Project administration, Supervision, Validation, Visualization, Writing – original draft, Writing – review and editing; Joey B Truong, Adriana Aguilar-Maldonado, Vinit Adani, Investigation; Monika Dzieciatkowska, Investigation, Methodology; Samara L Reck-Peterson, Conceptualization, Funding acquisition, Project administration, Supervision, Writing – original draft, Writing – review and editing

### Author ORCIDs

Jenna R Christensen ⓘ http://orcid.org/0000-0003-0323-6169
Agnieszka A Kendrick ⓘ http://orcid.org/0000-0003-3254-4582
Adriana Aguilar-Maldonado ⓘ http://orcid.org/0000-0003-1695-1719
Samara L Reck-Peterson ⓘ http://orcid.org/0000-0002-1553-465X

### Decision letter and Author response

Decision letter https://doi.org/10.7554/eLife.74538.sa1
Author response https://doi.org/10.7554/eLife.74538.sa2

## Additional files

### Supplementary files

• Supplementary file 1. BioID2 mass spectrometry datasets. All BioID2 mass spectrometry data referenced in the manuscript in *Figure 1C*, *Figure 1—figure supplement 1B, C*, *Figures 2A, 5A and 6A*, and *Figure 6—figure supplement 1A*. The first five tabs correspond to the FHIP1A, FHIP1B, FHIP2A, FHIP2B, and Hook2 carboxy-terminal BioID2 datasets from this study. The sixth and seventh tabs are the Hook1 and Hook3 carboxy-terminal datasets from *Redwine et al., 2017* for comparison.

• Supplementary file 2. Gene ontology (GO) analysis for FHIP BioID2 datasets. The results of GO searches using GOrilla (*Eden et al., 2009*; *Eden et al., 2007*). Enriched GO terms were identified by using the *Homo sapiens* database and by comparing two unranked lists of genes, using any significant hits in each dataset as the 'Target set' and any nonsignificant hits in the same dataset as the 'Background set' and searching for GO terms for process, function, and component. Standard search parameters (p value threshold of $10^{-3}$) were used. Each tab corresponds to the component, function, or process search results for one FHIP carboxy-terminal dataset.

• Supplementary file 3. Comparison between the different FHIP and Hook BioID2 mass

spectrometry datasets. The first tab corresponds to the comparison between FHIP1A, FHIP1B, FHIP2A, and FHIP2B carboxy-terminal BioID2 datasets shown in a Venn diagram in *Figure 1—figure supplement 1B*. The following four tabs correspond to the comparison between individual FHIP carboxy-terminal BioID2 datasets with Hook carboxy-terminal BioID2 datasets shown in *Figure 1— figure supplement 2*. The Hook1 and Hook3 carboxy-terminal datasets are from *Redwine et al., 2017*.

• Transparent reporting form

### Data availability
Supplementary files 1-3 contain all of the mass spectrometry data.

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
