## [Editor Report]

The microtubule motor cytoplasmic dynein-1 transports diverse membrane-bound organelles, but in most cases the mechanism of cargo recognition is unknown. Christensen, Kendrick, and colleagues use BioID, in vitro assays, and live-cell fluorescence imaging to show that the three Hook family cargo adaptors form complexes of distinct composition with proteins of the FHIP family. They map how specific FHIP proteins recruit dynein to different endosome and Golgi compartments. This study provides evidence for a new mechanism through which activation of dynein motility is coupled to the selection of cargo.

---

## [Decision Letter]

**Decision letter after peer review:**

Thank you for submitting your article "Cytoplasmic dynein-1 cargo diversity is mediated by the combinatorial assembly of FTS-Hook-FHIP complexes" for consideration by *eLife*. Your article has been reviewed by 3 peer reviewers, including Andrew Carter as Reviewing Editor and Reviewer #1, and the evaluation has been overseen by Anna Akhmanova as the Senior Editor. The reviewers have opted to remain anonymous.

Essential revisions:

From the reviewers discussions it was felt that determining whether Hook2/3 are present as single molecules (Figure 3) – reviewer #1 – would be a valuable addition to the manuscript.

*Reviewer #1:*

A key question in the cytoskeleton field is how the sole cytoplasmic dynein-1 motor is specifically recruited to the diverse cargoes it transports within the cell. In this manuscript, Christensen et al., provide a novel mechanism by which dynein's cargo specificity can be achieved. Using the cargo adaptor complex Fts-Hook-FHIP as a model system, the authors employ a BioID approach to determine the interactome of all FHIPs. They identify that FHIP1B and FHIP2A preferentially bind Hook1/Hook3 and Hook2, respectively. Addition of either Fts-Hook3-FHIP1B or Fts-Hook2-FHIP2A activates dynein in motility assays in vitro. Importantly, these distinct FHF complexes also associate with non-overlapping, dynein-driven motile cargo: FHIP1B binds endosomal proteins whereas FHIP2A acts in ER-to-Golgi Rab1-tubule transport.

Strengths

This manuscript reads well, and the claims are justified by mostly well-controlled experiments. Analysis of top BioID hits is followed up by an extensive array of validation techniques, including immunoprecipitation, live cell imaging and direct binding in vitro. Together, this makes a convincing case for FHIP1B and FHIP2A's preferential binding for different Hooks and cargo proteins. In addition, they show that Hook2 can activate dynein, adding to a growing list of dynein activating adaptors. The datasets generated here will also be a very useful resource for scientists looking to investigate the molecular basis for other adaptor-cargo links revealed by this study.

A weakness of the study

The authors conclude that Fts and FHIP do not/minimally alter properties of dynein movement. However, this is based on the comparison of full-length Hook and FHF – and we note that the velocities in these experiments are slower than previously reported. We also question whether full-length Hook in the absence of Fts-FHIP behaves as a single molecule.

In lines 293-294 the authors state: "The colocalization of these FHF complexes with moving dynein/dynactin had no effect or only minimal effects on dynein's motile properties including velocity, landing rates, and pausing frequencies." To ensure the validity of this conclusion, the authors may want to test whether all components in their TIRF assay are single molecules. Our own preparations of full-length Hook2 and Hook3 have consistently eluted in the void volume of gel filtration columns, showing a propensity to aggregate. We suggest the authors revise their single molecule experiments by following one or more of these suggestions:

1) Performing a sizing step as the last stage of their Hook purifications (SEC/or centrifugal gradients).

2) Tagging Hook with a SNAP or Halo tag to allow labelling and intensity/co-localisation measurements of this adaptor in TIRF.

3) Co-expressing Fts, Hook and FHIP. This helped us circumvent the heterogeneity we were observing with isolated full-length Hook, giving monodisperse and stoichiometric FHF complex. This has the added advantage of minimising the addition of Fts-FHIP sub-complexes to the TIRF assay.

The average velocities also appear lower than previously published work with truncated and full-length adaptors (Kendrick et al., 2019; Urnavicius et al., 2018). Can the authors explain the reasons why? Are these differences significant? In addition to the revisions suggested above, the authors should consider including a positive control (e.g. DDtruncatedHook3) to allow a direct comparison within this study.

Lines 296-298 state that: "Although run lengths increased slightly for FTS, Hook2, FHIP2A complexes (Figure 3 —figure supplement 2E), run lengths for FTS, Hook3, FHIP1B complexes remained unchanged (Figure 3 —figure supplement 3F)." However, in my view these conditions cannot be compared as the Hook3 experiments have omitted Lis1. Based on previous results and our attempts, Lis1 increases complex formation and is also likely to have an impact on the numbers of motors recruited. Adding Lis1 across all samples will allow a more robust comparison of velocity, run lengths and landing rate between Hook2 and Hook3 containing dynein complexes.

*Reviewer #2:*

In this manuscript, Christensen, Kendrick et al. investigate dynein-mediated intracellular transport of various cargos via different combinations of Hook and FHIP proteins. By applying proximity biotinylation (BioID) of FHIP in combination with mass spectrometry, they find that FHIP proteins selectively interact with marker proteins of different membrane-bound intracellular compartments. In particular, the authors highlight the following interactions: (1) FHIP1A and FHIP1B interact with endosomal marker proteins, (2) FHIP2A interacts with Golgi markers, and (3) FHIP2B interacts with both Golgi and mitochondrial marker proteins.

They further validate selectivity for interactions between FHIP and Hook proteins by co- immunoprecipitation from mammalian cell lines and focus on the partner complexes of: (1) Hook3 and FHIP1B and (2) Hook 2 and FHIP2A. The validate these interactions through co-immunoprecipitation assays and live-cell imaging. These experiments indicate that: (1) FHIP1B interacts with GTP-bound Rab5 to mediate the transport of early endosomes and (2) FHIP2A colocalizes with Rab1A-bound ER-to-Golgi tubular intermediates. Interestingly, co-immunoprecipitations do not show an interaction between FHIP2A and Rab1A. In addition, consistent with the idea that FHIPs confer cargo specificity, in in vitro reconstitution of motility experiments, FHIPs colocalize with motile dynein complexes, but do not alter motor properties like speed and landing rate.

This paper provides an important finding that combinations of adapters function to selectively load different vesicular cargos onto dynein motors. The BioID datasets provide a valuable resource for further studies that will broaden the repertoire of membrane-bound intracellular compartments and dynein adapters, opening the door to many future investigations. The data support the conclusions of the paper and are technically rigorous.

Major/conceptual points:

1) In the Figure 3 kymographs, the addition of FTS and FHIPs looks like it increases the number of bidirectional (jiggly) runs. Please quantify number per length of microtubule and run length. If not, that is fine. If so, it might indicate a more nuanced cargo loading effect.

2) In Figure 4C-F, please show endogenous FHIP staining or rationale why it is not shown. The authors state in the same paragraph that they have validated antibodies (by Western blot, in Figure 4B). I assume they do not work well for staining. If so, that is fine, but please add one sentence that using tagged proteins is not as reliable as staining for endogenous localization due to possible overexpression (and therefore sometimes mislocalization).

3) In the schematic in Figure 8, I anticipate that some readers will be confused by the placement of the centrosome in between the ER and Golgi, because generally we think of ER to Golgi transport as progressing from proximal to distal (relative to the nucleus). Since the centrosome is perinuclear, it often looks like it's on the nucleus in a max projection / 2D image, but I understand this might be confusing in a cartoon (which could be misinterpreted as being inside the nucleus).

*Reviewer #3:*

Recent years have seen rapid progress in our understanding of how autoinhibited cytoplasmic dynein 1 (dynein) is turned into a highly processive motor. A central role is played by cargo adaptor proteins, which bring dynein together with its activator dynactin. What is much less understood is how these so-called activating adaptors link the dynein-dynactin transport machine to its diverse cargo. Christensen, Kendrick, and colleagues provide insight into this problem by characterizing FHIP proteins, which are known to form "FHF" complexes with adaptors of the Hook family and a protein called FTS. To understand the cellular roles of the 4 human FHIP paralogs, the authors first determine the FHIP interactomes using proximity biotinylation in HEK293 cells. Proteomic analysis reveals that FHIP1A/B preferentially associate with endosomal proteins, while FHIP2A/B preferentially associate with proteins implicated in Golgi-related processes. Moreover, the interactome analysis suggests specific pairings exist between FHIP proteins and Hook adaptors, which is confirmed by co-immunoprecipitation experiments: FHIP1A/B form FHF complexes with Hook1 and 3, while FHIP2A interacts predominantly with Hook2. in vitro reconstitution of dynein-dynactin-FHF complexes and single-molecule motility assays with fluorescently labelled dynein and FHIP establish that FHIP2A-Hook2 and FHIP1B-Hook3 co-localize with processively moving dynein on microtubules, and that the presence of FHIP/FTS does not appreciably alter dynein's motile properties. To examine the role of FHIP1B and FHIP2A in cargo transport, the endogenous proteins are knocked out in U2OS cells and replaced by transgene-encoded fluorescent versions, which are imaged by spinning disk confocal microscopy. This shows that (overexpressed) FHIP1B and FHIP2A are present on motile compartments with a punctate and tubular morphology, respectively. FHIP1B (but not FHIP2A) co-localizes with Rab5A/B, co-immunoprecipitates with GTP-bound ("active") Rab5B, and does not associate with any of the 12 other Rab family members that are tested. This reveals FHIP1B as a Rab5-specific effector. On the other hand, co-localization experiments with VSV-G and proteins from the BioID dataset suggest that FHIP2A is involved in ER-to-Golgi transport. A visual screen of small GTPases associated with the ER or Golgi shows that Rab1A co-localizes with FHIP2A and Hook2. The authors do not find biochemical evidence that FHIP2A and Rab1A associate with one another, but two experiments suggest a functional link: overexpression of GDP-bound Rab1A decreases the number of mobile tubules marked by Rab1A or FHIP2A, and overexpression of transgene-encoded FHIP2A in the FHIP2A knock-out cell line rescues the decrease in motile Rab1A-marked tubules. The authors conclude that an FHF complex containing FHIP2A and Hook2 recruits dynein to ER-to-Golgi tubular transport intermediates.

This is a study of high technical quality that offers significant novel insight into the composition and function of FHF complexes. Through complementary approaches, the authors convincingly demonstrate that FHF complexes of distinct composition recruit dynein to specific cargo via membrane-bound Rab GTPases. These results therefore help explain how the single dynein motor can be selectively recruited to so many different types of cargo, which is an important unresolved question. The study focuses mostly on FHIP1B and FHIP2A, which leaves functional characterization of the other two FHIP proteins for the future. Also, while centrosomal clustering of Rab5-marked organelles in cells overexpressing FHIP1B suggests that FHIP1B promotes dynein-mediated early endosome transport, whether FHIP1B loss affects early endosome distribution has not been examined. Thus, while co-localization and biochemical data demonstrate that FHIP1B is a Rab5 effector, the functional implications of this interaction remain to be fully explored.

Given the already large number of technically sound experiments that support the authors' conclusions, I only have a single suggestion for an additional experiment.

Suggested (optional) experiment:

Examine the distribution/motility of Rab5-marked cargo in FHIP1B knock-out cells. This would complement the corresponding analysis of Rab1A-marked cargo in FHIP2A knock-out cells and would strengthen the functional link between FHIP1B and early endosome transport.

---

## [Author Response]

Reviewer #1:[…] In lines 293-294 the authors state: "The colocalization of these FHF complexes with moving dynein/dynactin had no effect or only minimal effects on dynein's motile properties including velocity, landing rates, and pausing frequencies." To ensure the validity of this conclusion, the authors may want to test whether all components in their TIRF assay are single molecules. Our own preparations of full-length Hook2 and Hook3 have consistently eluted in the void volume of gel filtration columns, showing a propensity to aggregate. We suggest the authors revise their single molecule experiments by following one or more of these suggestions:1) Performing a sizing step as the last stage of their Hook purifications (SEC/or centrifugal gradients).2) Tagging Hook with a SNAP or Halo tag to allow labelling and intensity/co-localisation measurements of this adaptor in TIRF.3) Co-expressing Fts, Hook and FHIP. This helped us circumvent the heterogeneity we were observing with isolated full-length Hook, giving monodisperse and stoichiometric FHF complex. This has the added advantage of minimising the addition of Fts-FHIP sub-complexes to the TIRF assay.

Our intention with the TIRF experiments was to establish minimal components required for complex formation and motility. Although we agree with the reviewer that the suggested experiments might shed additional light on the motile properties of these complexes, in our view these experiments are more suitable for a detailed follow-up study to identify the stoichiometry and molecular interactions between the components of these complexes. Performing these experiments for the current manuscript would be a substantial undertaking, as it will require us to re-express and purify Hook2 and Hook3 proteins and/or clone, co-express and co-purify FTS, FHIP and Hook proteins, as well as repeat all of the motility assays described in Figure 3. These experiments would significantly extend our revision timeline beyond the suggested two months and would not change the main conclusions of our work, which shows that different FHF complexes form in cells and in vitro and that dynein, dynactin, FTS and the different Hook and FHIP proteins are sufficient to form motile complexes. We added a phrase in the discussion that determining the precise stoichiometries of these complexes is an important future direction:

“An important future direction will be to determine the stoichiometries of these complexes and if the addition of cargo to moving dynein/dynactin/FHF complexes affects dynein’s motile properties.”

The average velocities also appear lower than previously published work with truncated and full-length adaptors (Kendrick et al., 2019; Urnavicius et al., 2018). Can the authors explain the reasons why? Are these differences significant? In addition to the revisions suggested above, the authors should consider including a positive control (e.g. DDtruncatedHook3) to allow a direct comparison within this study.

We agree that the velocities we describe appear slower than previously reported values for complexes containing full length and truncated Hook3. The most obvious difference is that the previous studies (McKenney et al., 2014, Urnavicius et al., 2018, Kendrick et al., 2019) used different experimental conditions compared to the current study.

Lines 296-298 state that: "Although run lengths increased slightly for FTS, Hook2, FHIP2A complexes (Figure 3 —figure supplement 2E), run lengths for FTS, Hook3, FHIP1B complexes remained unchanged (Figure 3 —figure supplement 3F)." However, in my view these conditions cannot be compared as the Hook3 experiments have omitted Lis1. Based on previous results and our attempts, Lis1 increases complex formation and is also likely to have an impact on the numbers of motors recruited. Adding Lis1 across all samples will allow a more robust comparison of velocity, run lengths and landing rate between Hook2 and Hook3 containing dynein complexes.

We realize that the motile properties of these complexes cannot be directly compared due to the addition of Lis1 to reaction mixtures containing Hook2. It was not our intention to make this comparison and we apologize for the confusion. We have clarified this in the text:

“Run lengths increased slightly for FTS, Hook2, FHIP2A complexes compared to complexes lacking FTS and FHIP2A (Figure 3 —figure supplement 2E). No difference in run lengths was detected for FTS, Hook3, FHIP1B complexes compared to those lacking FTS and FHIP1B (Figure 3 —figure supplement 3F).”

Reviewer #2:[…]Major/conceptual points:1) In the Figure 3 kymographs, the addition of FTS and FHIPs looks like it increases the number of bidirectional (jiggly) runs. Please quantify number per length of microtubule and run length. If not, that is fine. If so, it might indicate a more nuanced cargo loading effect.

Based on our analysis criteria, events referenced as “bidirectional (jiggly)” by the reviewer are considered diffusive (at least one bidirectional movement greater than 600 nm in each direction) and do not significantly change between experiments performed in the presence or absence of FTS and FHIP proteins. We added new panels (H and G) to Figure 3 —figure supplement 2 showing the number of motile, diffusive, and static events in single-molecule motility assays performed in the presence or absence of FTS and FHIP proteins. We added the following text to the Results section:

“The colocalization of these FHF complexes with moving dynein/dynactin had no effect or only minimal effects on dynein’s motile properties including velocity, landing rates, pausing frequencies, and number of processive runs (Figure 3D – G, Figure 3 —figure supplement 2C – H).”

2) In Figure 4C-F, please show endogenous FHIP staining or rationale why it is not shown. The authors state in the same paragraph that they have validated antibodies (by Western blot, in Figure 4B). I assume they do not work well for staining. If so, that is fine, but please add one sentence that using tagged proteins is not as reliable as staining for endogenous localization due to possible overexpression (and therefore sometimes mislocalization).

As the reviewer alluded to, the FHIP1B and FHIP2A antibodies do not work well for staining in our hands. As suggested, we have added the following to the text:

“However, the FHIP1B and FHIP2A antibodies we tested did not work well for immunofluorescence experiments in our hands.”.

We also clarified the wording in the prior sentence to “western blot-compatible antibodies.”

3) In the schematic in Figure 8, I anticipate that some readers will be confused by the placement of the centrosome in between the ER and Golgi, because generally we think of ER to Golgi transport as progressing from proximal to distal (relative to the nucleus). Since the centrosome is perinuclear, it often looks like it's on the nucleus in a max projection / 2D image, but I understand this might be confusing in a cartoon (which could be misinterpreted as being inside the nucleus).

We agree and considered several orientations for the organelles in Figure 8. One of the other difficulties is that though we may think of ER to Golgi transport as progressing from proximal to distal, relative to the nucleus, dynein transport is typically considered to proceed in the opposite direction, toward the nucleus. The movements happening in a cell in 3D are spatially complicated, making constructing a 2D image difficult. If there are any specific suggestions of how to change the cartoon accounting for all of these concerns, we are happy to edit it.

Reviewer #3:[…]Suggested (optional) experiment:Examine the distribution/motility of Rab5-marked cargo in FHIP1B knock-out cells. This would complement the corresponding analysis of Rab1A-marked cargo in FHIP2A knock-out cells and would strengthen the functional link between FHIP1B and early endosome transport.

We agree with the reviewer that this would be an ideal experiment to perform, and we attempted to quantify endosome motility in these cells via particle tracking. However, the large number of Rab5-tagged cargos made any type of rigorous analysis of motility extremely difficult. Additionally, Rab5-tagged endosomes are known to associate with multiple motors, potentially complicating this type of analysis. We chose to quantify fluorescence intensity at the centrosome as it was (1) a clearly observed phenotype by eye and (2) specifically focused on dynein-mediated transport. We suspect that there is no difference in the fluorescence intensity of Rab5-marked cargo at the centrosome between WT and *fhip2A* KO cells because there are very few endosomes at the centrosome in wild-type cells, making it difficult to observe any type of decrease when *fhip2A* is knocked out.